# Mitochondrial calcium exchange links metabolism with the epigenome to control cellular differentiation

Alyssa A. Lombardi[1], Andrew A. Gibb[1], Ehtesham Arif[1], Devin W. Kolmetzky[1], Dhanendra Tomar [1], Timothy S. Luongo[1], Pooja Jadiya [1], Emma K. Murray[1], Pawel K. Lorkiewicz [2], György Hajnóczky [3], Elizabeth Murphy[4], Zoltan P. Arany[5], Daniel P. Kelly[5], Kenneth B. Margulies[5], Bradford G. Hill[2] & John W. Elrod [1]

Fibroblast to myofibroblast differentiation is crucial for the initial healing response but excessive myofibroblast activation leads to pathological fibrosis. Therefore, it is imperative to understand the mechanisms underlying myofibroblast formation. Here we report that mito-chondrial calcium ($_mCa^{2+}$) signaling is a regulatory mechanism in myofibroblast differ-entiation and fibrosis. We demonstrate that fibrotic signaling alters gating of the mitochondrial calcium uniporter (mtCU) in a MICU1-dependent fashion to reduce $_mCa^{2+}$ uptake and induce coordinated changes in metabolism, i.e., increased glycolysis feeding anabolic pathways and glutaminolysis yielding increased α-ketoglutarate (αKG) bioavail-ability. $_mCa^{2+}$-dependent metabolic reprogramming leads to the activation of αKG-dependent histone demethylases, enhancing chromatin accessibility in loci specific to the myofibroblast gene program, resulting in differentiation. Our results uncover an important role for the mtCU beyond metabolic regulation and cell death and demonstrate that $_mCa^{2+}$ signaling regulates the epigenome to influence cellular differentiation.

[1] Center for Translational Medicine, Lewis Katz School of Medicine at Temple University, Philadelphia, PA 19140, USA. [2] Department of Medicine, Institute of Molecular Cardiology, Diabetes and Obesity Center, University of Louisville, Louisville, KY 40202, USA. [3] Department of Pathology Anatomy and Cell Biology, MitoCare Center for Mitochondrial Imaging Research and Diagnostics, Thomas Jefferson University, Philadelphia, PA 19107, USA. [4] Systems Biology Center, National Heart Lung and Blood Institute, Bethesda, MD 20892, USA. [5] Translational Research Center, Perelman School of Medicine, University of Pennsylvania, Philadelphia, PA 19014, USA. Correspondence and requests for materials should be addressed to J.W.E. (email: elrod@temple.edu)

Fibroblast to myofibroblast differentiation is a universal response to injury whereby fibroblasts differentiate from a quiescent structural role into contractile and synthetic myofibroblasts, which are vital to wound healing[1–3]. However, the reparative characteristics of myofibroblasts also contribute to pathological fibrosis. These cells produce copious extracellular matrix (ECM) proteins such as periostin, collagen, and fibronectin, and can remodel tissues due to de novo expression of α-smooth muscle actin (α-SMA)[3,4]. Differentiation of fibroblasts into myofibroblasts is initiated and sustained by several agonists including: Transforming Growth Factor-beta (TGFβ), Angiotensin II (AngII), Endothelin 1 (ET1), and mechanical tension, which initiate distinct, yet interconnected, signaling pathways[1,5]. Since acquisition and retention of the myofibroblast state is an important mediator of pathological fibrosis, it is important to define the molecular mechanisms that mediate this differentiation process[3,5–7].

Recently it has become appreciated that a sustained elevation in cytosolic calcium ($_cCa^{2+}$) promotes the conversion of quiescent fibroblasts into myofibroblasts. Profibrotic mediators TGFβ, AngII, and ET1 trigger an increase in $_cCa^{2+}$[8–10]. In addition, multiple groups have established transient receptor potential (TRP) channels as contributors to cardiac fibrosis and myofibroblast differentiation. TRPV4, TRPM7, and TRPC6 have all been implicated in myofibroblast differentiation[11–13]. While $_cCa^{2+}$ signaling appears to be necessary for both TGFβ-dependent and TGFβ-independent signaling pathways, other cellular Ca²⁺ domains, such as mitochondrial calcium ($_mCa^{2+}$), have not been explored. Elevations in $_cCa^{2+}$ are rapidly integrated into mitochondria through the mitochondrial calcium uniporter channel complex (mtCU) due to the high electromotive force generated by the electron transport chain (ETC) (Δψ = ~ −160 mv)[14]. $_mCa^{2+}$ directly impacts cellular bioenergetics through the activation of dehydrogenases in the TCA cycle and by modulating ETC function[15–17]. This is intriguing as alterations in metabolism are reported to be essential to cell fate determination, i.e., pluripotency vs. committed/specified cells[18–20]. Indeed, alterations in the levels of various metabolites have been linked to the activity of epigenetic-modifying enzymes, providing a direct link between cellular metabolism and gene expression[18,20].

Here, we investigate the role of $_mCa^{2+}$ uptake in cellular differentiation. This study reveals that alterations in $_mCa^{2+}$ exchange, via MICU1-dependent mtCU gating, is a central regulatory mechanism linking canonical signaling pathways with adaptive changes in mitochondrial metabolism and epigenetics that are necessary to drive cellular differentiation.

## Results

**Ablation of $_mCa^{2+}$ uptake in fibroblasts.** To examine the contribution of $_mCa^{2+}$ uptake to myofibroblast differentiation, we conditionally deleted *Mcu*, the pore-forming subunit of the mtCU that is necessary for $_mCa^{2+}$ uptake (Fig. 1a)[17,21–23]. Mouse embryonic fibroblasts (MEFs) were isolated from E13.5 *Mcu*$^{fl/fl}$ embryos and transduced with adenovirus-encoding Cre recombinase (Ad-Cre) or beta-galactosidase (Ad-βgal, adenoviral control) for 24 h, and 4 days later cell lysates were analyzed by Western blot. Cre-mediated deletion of exons 5–6 caused complete loss of MCU protein (Fig. 1c). We also observed a loss of mtCU components MCUB and EMRE (Fig. 1c), likely attributed to protease mediated degradation of the other structural/channel-forming mtCU components[24]. Voltage-dependent anion channel (VDAC) and the UQCRC2 (Ubiquinol-cytochrome-c reductase complex core protein 2) subunit of Complex III (CIII) were used as mitochondrial loading controls and tubulin served as a total

lysate loading control. Next, *Mcu*$^{fl/fl}$ MEFs were infected with Ad-Cre or Ad-βgal and 72 h later transduced with adenovirus encoding a mitochondrial-targeted genetic Ca²⁺ reporter (Mito-R-GECO) for 48 h. Prior to live-cell imaging, cells were loaded with the calcium sensitive dye Fluo-4 AM to measure cytosolic calcium ($_cCa^{2+}$) transients. After baseline recordings, cells were treated with ATP to initiate purinergic receptor-mediated IP3R Ca²⁺ release. Control MEFs (Ad-βgal) displayed robust $_mCa^{2+}$ transients, whereas *Mcu*$^{−/−}$ MEFs (Ad-Cre) displayed complete loss of $_mCa^{2+}$ uptake (Fig. 1d, e). Further, loss of MCU-mediated uptake elicited a significant increase in $_cCa^{2+}$ transients, suggesting that mitochondria buffer $_cCa^{2+}$ signaling in fibroblasts (Fig. 1f, g). In addition, loss of $_mCa^{2+}$ uptake enhanced cytosolic signaling. Using an adenovirus-encoding NFATc1-GFP, we measured nuclear translocation of NFATc1 following fibrotic stimuli. NFATc1 normally resides in the cytoplasm, but upon increased $_cCa^{2+}$ NFATc1 is dephosphorylated and translocates into the nucleus to regulate gene transcription[25]. Treatment with TGFβ or AngII for 24 h induced nuclear translocation of NFATc1 in control cells (Ad-βgal) and this was slightly potentiated in *Mcu*$^{−/−}$ fibroblasts (Ad-Cre) (Supplementary Fig. 1a, b).

**Loss of $_mCa^{2+}$ uptake promotes myofibroblast differentiation.** To determine the role of $_mCa^{2+}$ signaling in myofibroblast differentiation, *Mcu*$^{fl/fl}$ MEFs were infected with Ad-Cre or Ad-βgal and 5 days later treated with TGFβ or AngII. MEFs were examined for differentiation into a myofibroblast by quantifying α-smooth muscle actin (α-SMA) stress fiber formation, the prototypical marker of myofibroblasts[3]. *Mcu*$^{−/−}$ MEFs (Ad-Cre) displayed increased myofibroblast formation at baseline (vehicle) and following 24 h TGFβ or AngII treatment as evidenced by an increase in the percentage of α-SMA+ cells and a ~4-fold increase in α-SMA expression vs. controls (Ad-βgal) (Fig. 1h–l). Functionally, *Mcu*$^{−/−}$ MEFs displayed increased contraction of collagen gel matrices, even without TGFβ or AngII treatment, indicative of enhanced acquisition of the myofibroblast phenotype (Fig. 1m, n). We also observed that loss of mtCU-mediated Ca²⁺-uptake alone was sufficient to increase expression of key myofibroblast genes including: collagens (*Col1a1* and *Col3a1*), α-SMA (*Acta2*), periostin (*Postn*), fibronectin (*Fn1*) and platelet derived growth factor receptor alpha (*Pdgfra*) (Fig. 1o). Importantly, the observed enhancement in *Mcu*$^{−/−}$ α-SMA+ cells and gel contraction was not due to increased proliferation. *Mcu*$^{−/−}$ MEFs showed significantly reduced proliferation rates, as measured by DNA content, which is also characteristic of a more differentiated cell type (Fig. 1p). Collectively, these data show that loss of $_mCa^{2+}$ uptake promotes myofibroblast differentiation.

**Fibrotic stimuli alter mtCU gating to reduce $_mCa^{2+}$ uptake.** Given the significant impact that loss of $_mCa^{2+}$ uptake had on myofibroblast formation we next examined if acute fibrotic signaling directly altered mtCU function. First, we used live cell imaging to measure Ca²⁺ transients in WT fibroblasts incubated with or without TGFβ for 12 h. MEFs were transduced with Mito-R-GECO for 48 h and loaded with Fura-2 prior to imaging to measure $_mCa^{2+}$ and $_cCa^{2+}$ transients in response to ATP-stimulated IP3R Ca²⁺ release. TGFβ pretreated cells showed significantly increased $_cCa^{2+}$ and significantly decreased total $_mCa^{2+}$ load (Fig. 2a, b, Supplementary Fig. 2a, b).

Next, we measured the impact of acute TGFβ signaling on $_mCa^{2+}$ uptake in a permeabilized cell system. This high-fidelity system allows careful monitoring of uptake independent of changes in other calcium transport mechanisms. After treating WT MEFs with TGFβ for 12 h, fibroblasts were permeabilized

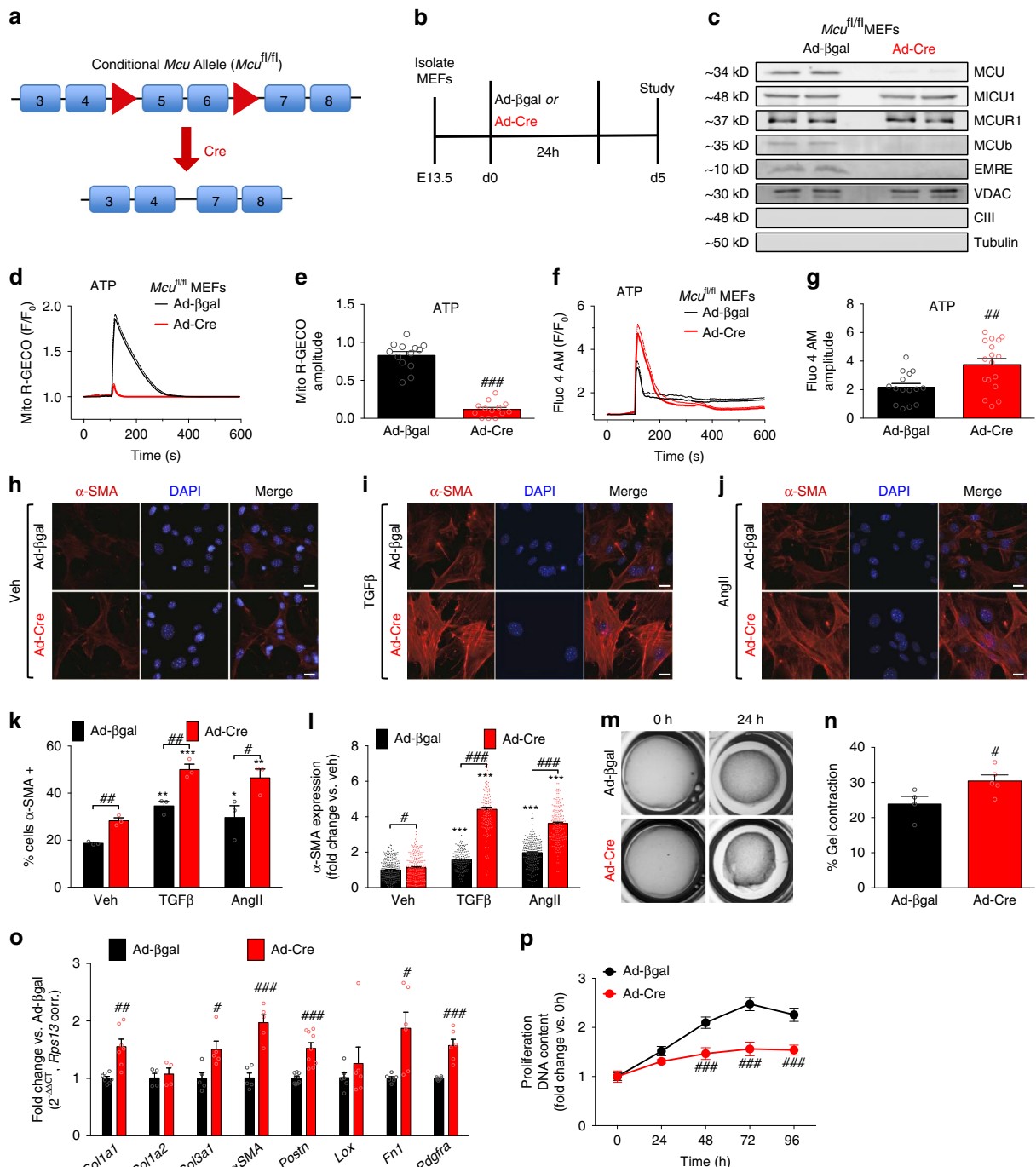

**Fig. 1** Loss of mCa2+ uptake enhances the myofibroblast differentiation. **a** *Mcu* conditional allele with LoxP sites flanking exons 5–6. **b** Experimental timeline for deletion of *Mcu* in mouse embryonic fibroblasts (MEFs). MEFs were isolated from *Mcu*fl/fl embryos at E13.5 and infected with adenovirus encoding Cre recombinase (Ad-Cre) or the control beta-galactosidase (Ad-βgal) for 24 h. **c** Expression of mtCU components was examined by Western blot in *Mcu*−/− (Ad-Cre) and control (Ad-βgal) MEFs. MICU1 Mitochondrial Ca2+ Uptake 1, MCUR1 Mitochondrial Ca2+ Uniporter Regulator 1, MCUB Mitochondrial Ca2+ Uniporter B, EMRE Essential MCU Regulator. Mitochondrial loading controls: Voltage-dependent anion channel (VDAC) and complex III (CIII, subunit-UQCRC2) served as mitochondrial loading controls and tubulin as total lysate loading control. **d** *Mcu*−/− and control were transduced with adenovirus encoding the mitochondrial calcium sensor, Mito R-GECO. 1 mM ATP was delivered to initiate IP3R-mediated Ca2+ release; n = 13 cells. **e** Amplitude (peak intensity–baseline). **f** *Mcu*−/− and control were loaded with the Ca2+-sensitive dye Fluo-4 AM. Fluorescence was recorded during 1 mM ATP treatment; n = 15 cells Ad-βgal, n = 17 cells Ad-Cre. **g** Amplitude (peak intensity–baseline). **h–l** MEFs were treated with TGFβ or AngII for 24 h and immunofluorescence was performed by costaining with α-smooth muscle actin (α-SMA) antibody (red) and DAPI (blue); n = 3. **h–j** Representative images. **k** Percentage of α-SMA positive cells. **l** α-SMA expression (fluorescence intensity). **m**, **n** Collagen gel contraction assay; n = 4 Ad-βgal, n = 5 Ad-Cre. **m** Representative images. **n** Gel contraction calculated as percent change from time 0 h. **o** Fold change in expression of myofibroblast genes (vs. Ad-βgal control). Col1a1 collagen type I alpha 1 chain, Col1a2 collagen type I alpha 2 chain, Col3a1 collagen type III alpha 1 chain, α-SMA (Acta2) α-smooth muscle actin, Postn periostin, Lox lysyl oxidase, Fn1 fibronectin 1, Pdgfra platelet derived growth factor receptor alpha col1a1 (n = 7), col1a2 (n = 4), col3a1 (n = 5), aSMA (n = 5), Postn (n = 9), Lox (n = 6), Fn1 (n = 6), Pdgfra (n = 6). **p** Cell proliferation measured by quantifying DNA content; n = 6. Ca2+ traces: solid line = mean, dashed line = SEM. Data shown as mean ± SEM. ***p < 0.001, **p < 0.01, *p < 0.05 vs. vehicle control analyzed by ANOVA. ###p < 0.001, ##p < 0.01, #p < 0.05 vs. Ad-βgal analyzed by t-test. Scale bar = 50 μm. Also see Supplementary Fig. 1

with digitonin, in the presence of thapsigargin (SERCA inhibitor to prevent ER $Ca^{2+}$ uptake) and CGP-37157 (NCLX inhibitor to block $_mCa^{2+}$ efflux), and loaded with the $Ca^{2+}$ sensor Fura-2 for ratiometric monitoring using a spectrofluorometer. An increase in Fura-2 signal signifies an increase in bath $Ca^{2+}$ and a decrease in Fura-2 signal after each bolus represents $_mCa^{2+}$ uptake. TGFβ-treated fibroblasts displayed a decrease in $_mCa^{2+}$ uptake following the delivery of ~0.25–2 μM $[Ca^{2+}]$ (representative trace shown in Fig. 2c). Importantly, simultaneous monitoring of mitochondrial membrane potential (Δψ) using the ratiometric reporter, JC-1, showed no difference in the driving force for uptake (Fig. 2d). After calibration of the Fura-2 reporter (Supplementary Fig. 2c), we quantified the percentage of $_mCa^{2+}$ uptake over a range of bath $Ca^{2+}$ concentrations and data points were fit to the Hill equation using a nonlinear least-squares fit. From the dose response curve, we observed the nonlinear nature of mtCU-mediated $_mCa^{2+}$ uptake, consistent with other reports (Fig. 2e)[26–28]. TGFβ treatment for 12 h shifted the dose-response curve to the right, demonstrating an increase in the $[Ca^{2+}]$ threshold for $_mCa^{2+}$ uptake (Fig. 2e, f). The calculated $K_d$ value was ~1.5 μM in control cells and ~1.9 μM in TGFβ-treated cells, indicating that following TGFβ higher $[_cCa^{2+}]$ was needed to achieve 50% maximal mtCU uptake (Fig. 2g). In addition, the Hill coefficient identified a difference in the slopes of the dose response curves (Fig. 2g), 4.29 in control cells vs. 10.27 in TGFβ-treated cells, demonstrating that TGFβ indeed enhanced mtCU gating, allowing virtually no uptake until a given threshold was reached.

To probe the mechanism responsible for TGFβ-induced alterations in $_mCa^{2+}$ uptake, we treated WT fibroblasts with TGFβ and 12, 24, 48, and 72 h later extracted protein to examine expression of mtCU components. Western blot analysis revealed a dramatic increase in MICU1 expression 12 h after treatment (Fig. 2h, i). CIII (subunit UQCRC2) and VDAC served as mitochondrial loading controls and tubulin served as a total lysate loading control. Since the MICU1/MCU ratio underlies tissue-specific differences in the mtCU $[_cCa^{2+}]$ threshold of uptake[29], we quantified the relative change in MICU1/MCU ratio. TGFβ treatment rapidly increased the MICU1/MCU ratio (Fig. 2j). We also observed a similarly large increase in MICU1 expression in MEFs treated with AngII (Fig. 2m–o), suggesting this is a conserved mtCU regulatory mechanism during myofibroblast differentiation. The substantial increase in the MICU/MCU ratio is in agreement with our observed change in $_mCa^{2+}$ uptake following TGFβ treatment and is consistent with other reports ascribing that MICU1 is a gatekeeper restricting mtCU-mediated $Ca^{2+}$ uptake at signaling levels of $[_cCa^{2+}]$[28,30]. We propose that profibrotic agonists signal to acutely upregulate MICU1 expression to inhibit $_mCa^{2+}$ uptake and initiate myofibroblast differentiation signaling. The relative expression of additional mtCU components was also quantified (Supplementary Fig. 2d–k). This increase in MICU1 protein expression is likely due to TGFβ-mediated and AngII-mediated transcriptional upregulation of Micu1 (Fig. 2k, p) that occurs as early as 1 h following stimulation. Neither TGFβ or AngII induced a change in MCU expression (Supplementary Fig. 2l, n). Increases in the MICU1/MCU ratio were also evident at the transcriptional level (Supplementary Fig. 2m, o). We found the same phenomenon in mouse adult cardiac fibroblasts (ACFs) treated with TGFβ and AngII-upregulation of MICU1 and an increase in the MICU1/MCU ratio (Fig. 2l, q, Supplementary Fig. 2p–s).

**TGFβ/AngII signaling elicits dynamic changes in fibroblast metabolism.** $_cCa^{2+}$ is integrated into the mitochondrial matrix via the mtCU, a mechanism theorized to integrate cellular demand with metabolism and respiration[17,31–33]. Further, metabolic reprogramming is required for numerous cellular differentiation programs[19,20] and recent studies suggest that enhanced glycolysis promotes fibroblast differentiation[34,35]. This prompted us to examine metabolic changes in glycolysis and oxidative phosphorylation during myofibroblast differentiation. $Mcu^{fl/fl}$ MEFs were transduced with Ad-Cre or Ad-βgal and 5 days later treated with TGFβ or AngII for 12, 24, 48, or 72 h, followed by measurement of extracellular acidification rates (ECAR, glycolysis) and oxygen consumption rates (OCR, OxPhos) using a Seahorse XF96 analyzer (Supplementary Fig. 3a–c). TGFβ elicited a significant increase in basal respiration (~135% increase from baseline) and glycolysis (>400% increase from baseline) peaking 48 h after treatment (Fig. 3a, c). AngII likewise caused a rapid increase in glycolysis (45% increase from baseline), peaking ~12 h; however, AngII caused a slight decrease in basal respiration (Fig. 3b, d). Interestingly, loss of MCU (Ad-Cre) further enhanced the increased glycolysis induced by both TGFβ and AngII >2-fold, as compared to control (Ad-βgal) (Fig. 3e). Other Seahorse metabolic parameters under all conditions are reported in Supplementary Fig. 3d–g.

Next, using a quantitative metabolomics approach, concentrations of fibroblast metabolites were quantified by mass spectrometry in $Mcu^{-/-}$ (Ad-Cre) and control (Ad-βgal) fibroblasts at baseline and 12 h post-TGFβ. These data supported that TGFβ-dependent changes in mtCU gating mediated the increase in glycolysis observed by Seahorse analysis. $Mcu^{-/-}$ MEFs (Ad-Cre) displayed higher levels of the glycolytic intermediates: glucose-6-phosphate (G-6-P), fructose-6-phosphate (F-6-P), fructose-1,6-bisphosphate (F-1,6-BP), glyceraldehyde-3-phosphate (GA3P), dihydroxyacetone phosphate (DHAP) and glycerol-3-phosphate (G-3-P) (Fig. 3f–m). Importantly, F-1,6-BP, the glycolytic intermediate produced in the first committed step of glycolysis, was significantly increased following TGFβ treatment and this increase was potentiated by loss of MCU (Ad-Cre) (Fig. 3i). F-1,6-BP is metabolized into GA3P and DHAP, and concentrations of these metabolites followed a similar trend with an increase post-TGFβ, which was similarly potentiated by loss the loss of $_mCa^{2+}$ uptake (Fig. 3j, l). In addition to energy production, glycolysis supplies metabolic intermediates for ancillary biosynthetic pathways necessary for cellular growth and differentiation. For example, the pentose phosphate pathway (PPP) generates ribulose-5-phosphate (Ru-5-P) along with NADPH, which are critical for nucleotide and fatty acid/phospholipid synthesis, respectively (Supplementary Fig. 4a)[36]. Following TGFβ, $Mcu^{-/-}$ MEFs exhibited increased levels of 6-phosphogluconate (6-PG), Ru-5-P, and ribose-5-phosphate (R-5-P) compared to vehicle treated controls (Supplementary Fig. 4b–d).

To determine the necessity of enhanced glycolytic flux on myofibroblast formation, we modulated a rate-limiting enzyme of glycolysis, phosphofructokinase 1 (PFK1). PFK1 is allosterically activated by fructose-2,6-bisphosphate (F-2,6-BP), the levels of which are regulated by the bifunctional enzyme phosphofructokinase 2 (PFK2)/fructose bisphosphatase 2 (FBP2) (Fig. 3f)[37]. Employing adenoviruses encoding a phosphatase-deficient PKF2/FBP2 mutant (S32A, H258A; Ad-Glyco-High) or kinase-deficient PFK2/FBP2 mutant (S32D, T55V; Ad-Glyco-Low) we examined the impact of modulating glycolytic rates during myofibroblast differentiation (Fig. 3n, o)[38,39]. The PFK2/FBP2 mutant adenoviruses also encoded GFP driven by a separate CMV promoter, allowing identification of transduced cells apart from uninfected fibroblasts. Ad-Glyco-High expression increased glycolysis in both control (Ad-βgal) and $Mcu^{-/-}$ (Ad-Cre) MEFs, while Ad-Glyco-Low expression inhibited the increased glycolysis observed in $Mcu^{-/-}$ MEFs (Fig. 3p). Control and $Mcu^{-/-}$ MEFs were infected with either Ad-Glyco-High or Ad-Glyco-Low and 24 h

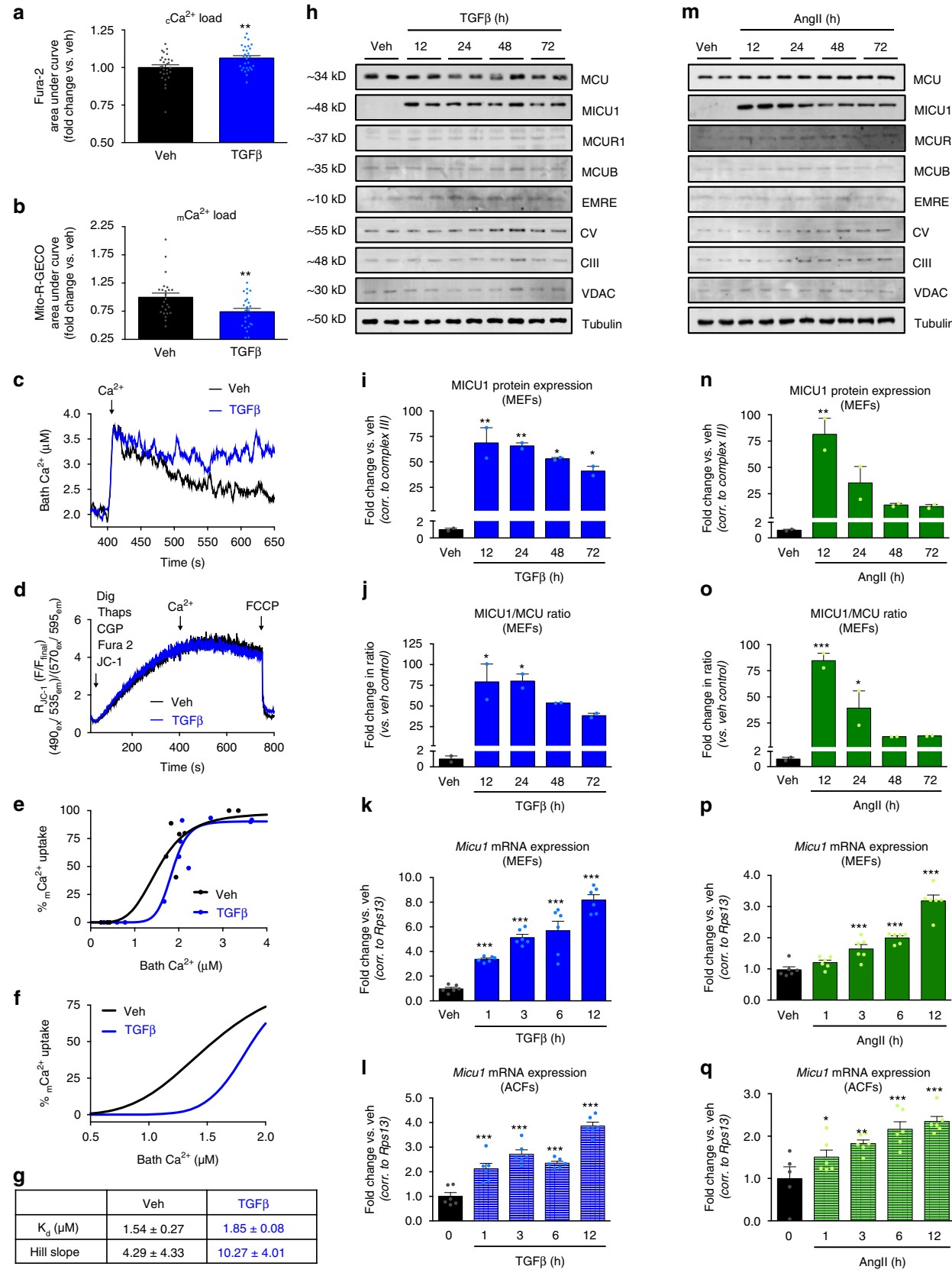

later treated with TGFβ or AngII for 24 h followed by quantification of α-SMA+ cells by immunofluorescence. Enhancing glycolysis was sufficient to drive myofibroblast formation (Fig. 3q, r) and potentiated cellular differentiation elicited by TGFβ and AngII (Fig. 3q, r). Inhibition of glycolysis (Ad-Glyco-Low) prevented TGFβ-mediated and AngII-mediated differentiation and reverted the enhanced differentiation in $Mcu^{-/-}$ fibroblasts back to control levels (Fig. 3s, t).

**Fig. 2** Profibrotic stimuli alter mtCU gating to reduce $_mCa^{2+}$ uptake. **a** MEFs plus or minus (+/−) 12 h TGFβ were loaded with $Ca^{2+}$-sensitive dye Fura-2. Fluorescence was recorded and 1 mM ATP was delivered to initiate IP3R-mediated $Ca^{2+}$ release. Cytosolic $Ca^{2+}$ ($_cCa^{2+}$) load was determined by calculating area-under-the-curve (AUC) of $_cCa^{2+}$ transients; $n = 28$ cells. **b** MEFs +/− 12 h TGFβ were transduced with adenovirus encoding the mitochondrial calcium ($_mCa^{2+}$) sensor, Mito-R-GECO. Fluorescence was recorded during 1 mM ATP treatment. $_mCa^{2+}$ load was determined by calculating AUC; $n = 24$ cells. **c–g** MEFs +/− 12 h TGFβ were permeabilized with digitonin in the presence of thapsigargin (SERCA inhibitor) and CGP-37157 (NCLX inhibitor) and loaded with $Ca^{2+}$ sensor Fura-2 and Δψ sensor JC-1 for ratiometric monitoring during $Ca^{2+}$ additions. **c** Representative $Ca^{2+}$ traces in untreated (black) and TGFβ-treated (blue) MEFs. **d** JC-1 derived Δψ in untreated (black) and TGFβ-treated (blue) MEFs. **e, f** Dose response curve of $_mCa^{2+}$ uptake following different [$Ca^{2+}$] boluses. **g** Kinetic parameters derived from data in panel **e**. **h–j** MEFs were treated with TGFβ and cell lysates were immunoblotted for components of the mtCU, including pore forming subunit MCU and regulatory subunits MICU1 (Mitochondrial $Ca^{2+}$ Uptake 1), MCUR1 (Mitochondrial $Ca^{2+}$ Uniporter Regulator 1), MCUB, and EMRE (Essential MCU Regulator), as well as OxPhos Complexes CV (ATP5A) and CIII (subunit-UQCRC2), VDAC (Voltage-dependent anion channel), and tubulin. Mitochondrial loading controls: VDAC and CIII; total lysate loading control: tubulin. Band density was normalized to CIII; $n = 3$. **k, l** MEFs and mouse adult cardiac fibroblasts (ACFs) were treated with TGFβ and *Micu1* mRNA was analyzed by qPCR. $n = 6$. **m–o** MEFs were treated with AngII and cell lysates were immunoblotted for MCU, MICU1, MCUR1, MCUB, and EMRE, as well as OxPhos Complexes CV and CIII, VDAC, and tubulin. Mitochondrial loading controls: VDAC and CIII. Total lysate loading control: tubulin. Band density was normalized to CIII; $n = 3$. **p, q** MEFs and mouse ACFs were treated with AngII and *Micu1* mRNA was analyzed by qPCR ($n = 6$). Data shown as mean ± SEM. ***$p < 0.001$, **$p < 0.01$, *$p < 0.05$ vs. vehicle control analyzed by ANOVA. Also see Supplementary Fig. 2

Next, we evaluated mitochondrial metabolism since it is well established that $_mCa^{2+}$ signaling directly impacts TCA cycle intermediates by the modulation of pyruvate dehydrogenase (PDH) and α-ketoglutarate dehydrogenase (αKGDH) activity. $_mCa^{2+}$ activates PDH phosphatase (PDP1), which dephosphorylates the PDH E1α subunit and thereby increases PDH activity to convert pyruvate to acetyl-CoA[40]. Western blot analysis of phosphorylated PDH (p-PDH E1α, inactive) revealed significantly increased p-PDH E1α/PDH in $Mcu^{-/-}$ MEFs (Ad-Cre) at baseline compared to controls (Ad-βgal) (Fig. 4b, c). Further, both TGFβ and AngII increased the ratio of p-PDH E1α/PDH, which was potentiated in $Mcu$-null fibroblasts (Fig. 4d). Accordingly, metabolomics analysis revealed that TGFβ increased pyruvate by ~60%, consistent with inhibition of PDH (Fig. 4e). $Mcu^{-/-}$ MEFs had increased pyruvate at baseline and following TGFβ compared to controls (Fig. 4e). Acetyl-CoA was decreased in $Mcu^{-/-}$ MEFs at baseline, also consistent with inactive PDH (Fig. 4f). Following TGFβ, acetyl-CoA increased in $Mcu^{-/-}$ MEFs, but did not change in control cells (Fig. 4f). Nonetheless, acetyl-CoA levels were 100 times lower than pyruvate levels, suggesting that pyruvate was not entering the TCA cycle via PDH. Consistent with reduced glucose/pyruvate oxidation, citrate levels were significantly reduced following TGFβ (Fig. 4g). Interestingly, TGFβ increased α-ketoglutarate (αKG (Fig. 4h). Further, $Mcu^{-/-}$ fibroblasts exhibited increased αKG at baseline and following treatment with TGFβ, as compared to controls (Fig. 4h). Other TCA cycle intermediates succinate, fumarate, and malate were unchanged by TGFβ or loss of MCU (Fig. 4i–k). Reduced glucose-dependent TCA flux increases anaplerotic elevations in αKG via glutaminolysis[41,42]. TGFβ decreased glutamine (Gln) and glutamate (Glu) levels in control cells and $Mcu^{-/-}$ fibroblasts displayed an increase in the αKG/Gln ratio at baseline and after TGFβ stimulation (Fig. 4l–n), suggesting that TGFβ activates glutaminolysis to increase cellular levels of αKG. All other metabolite concentrations are reported in Supplementary Fig. 5 and Supplementary Table 1.

To directly test the significance of glutamine utilization in myofibroblast differentiation, we diminished glutaminolysis in WT MEFs treated with or without TGFβ for 48 h and assessed α-SMA formation by immunofluorescence. Glutaminolysis was constrained by either removal of Gln from media (Fig. 4o–q) or treatment with CB-839 (Fig. 4r–t). CB-839 is a potent selective inhibitor of glutaminase 1 (GSL1), the enzyme which converts Gln into Glu in the first step of glutaminolysis (Fig. 4a)[43]. In both of these experiments myofibroblast differentiation was significantly attenuated, suggesting that myofibroblast differentiation is dependent on glutamine-derived αKG (Fig. 4o–t).

**Chromatin remodeling activates the myofibroblast gene program.** αKG is a cofactor for several dioxygenases, including the epigenetic modifiers ten-eleven translocation enzymes (TETs) and Jumonji-C (JmjC)-domain-containing demethylases (JmjC-KDMs), which demethylate DNA cytosine residues and histone lysine residues, respectively (Fig. 5a)[44]. We hypothesized that the observed increase in αKG following TGFβ or loss of MCU altered epigenetic signaling to promote myofibroblast differentiation. We assessed global DNA methylation by ELISA in $Mcu^{-/-}$ (Ad-Cre) and control (Ad-βgal) MEFs at baseline and following treatment with TGFβ. We observed slight, but non-significant, decreases in global DNA methylation with TGFβ and loss of MCU (Fig. 5b). Subsequently, $Mcu^{-/-}$ (Ad-Cre) and control (Ad-βgal) MEFs were treated with TGFβ and cell lysates were examined for histone 3 (H3) lysine (K) methylation at key residues regulated by JmjC-KDMs – H3K27, H3K9, and H3K4 (Fig. 5c). Fibroblasts treated with TGFβ exhibited a progressive decrease in dimethylation of H3K27 (H3K27me2) over time (Fig. 5c, d). $Mcu^{-/-}$ MEFs exhibited less dimethylation at baseline and post-TGFβ compared to controls (Fig. 5c, d). Quantification of other methylation residues is reported in Supplementary Fig. 6a–e. H3K27me2 is implicated in regulating cell fate by preventing inappropriate enhancer activation[45] and generally is associated with heterochromatin and gene suppression[46].

To directly examine the role of H3K27me2 in controlling the myofibroblast gene program, we immunoprecipitated chromatin using an H3K27me2-specific antibody and ChIP'd DNA was analyzed by qPCR in key regulatory promoter regions of *Postn* and *Pdgfra*, genes, which are early and robust indicators of fibroblast activation[4,47,48]. In control cells (Ad-βgal), H3K27me2 was enriched at the *Postn* and *Pdgfra* loci and these marks were lost after 12 h of TGFβ with a concordant increase in mRNA expression (Fig. 5e, f and Supplementary Fig. 6f, g). Furthermore, $Mcu^{-/-}$ MEFs (Ad-Cre) exhibited a lack of H3K27me2 enrichment at the *Postn* and *Pdgfra* promoters at baseline, which we hypothesize underlies the enhanced expression of these genes and suggests $Mcu^{-/-}$ cells are primed for myofibroblast formation (Fig. 5e, f and Supplementary Fig. 6f, g). Importantly, binding sites for transcription factors known to be prominent drivers of myofibroblast differentiation such as serum response factor (SRF), SMAD family member 3 (SMAD3), nuclear factor for activated T-cells (NFAT), and myocyte enhancer factor-2 (MEF2) were predicted by MatInspector to be flanked by, or in close approximation, to the regulatory regions probed by our qPCR primer sets (Fig. 5e and Supplementary Fig. 6f).

To further examine the role of TGFβ in transcriptional regulation and chromatin structure, we used the assay for

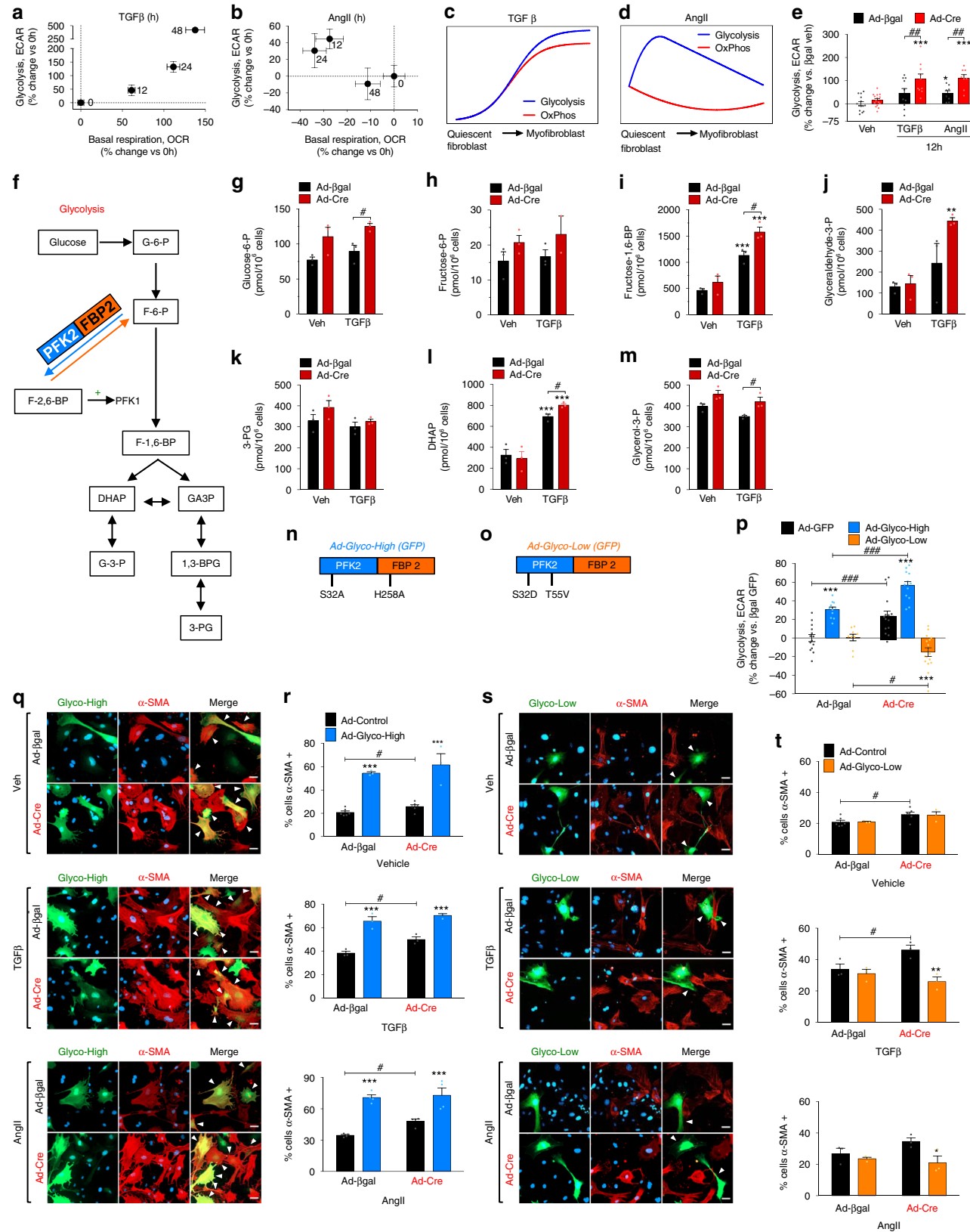

transposase-accessible chromatin utilizing deep-sequencing (ATAC-seq)[49], coupled with RNA sequencing (RNA-seq) in the same samples, to analyze chromatin accessibility and gene expression in control and TGFβ-treated MEFs, respectively. MEFs treated with TGFβ exhibited enhanced chromatin accessibility in the transcriptional regulatory region of key

myofibroblast genes including: *Postn* (Fig. 5g), *Acta2* (α-SMA), *Pdgfra*, *Col1a1*, *Lysyl Oxidase* (*Lox*), *Tgf*β, and *Wnt family member 1* (*Wnt1*) (Supplementary Fig. 7a–f). Importantly, ATAC-derived increases in open chromatin structure correlated with increased or decreased mRNA transcription in a directionality in agreement with what has been reported for myofibroblast

**Fig. 3** TGFβ/AngII signaling elicits rapid and dynamic changes in fibroblast metabolism. **a–e** MEFs were treated with fibrotic stimuli and a Seahorse XF96 analyzer measured extracellular acidification rates (ECAR, glycolysis) or oxygen consumption rates (OCR, OxPhos); $n =$ individual dots/group, 3 experiments per study. **a, b** Percent change in glycolysis (y-axis) vs. percent change in basal respiration (x-axis) following stimulation with TGFβ or AngII for 0, 12, 24, or 48 h. **c, d** Schematic representations of changes in glycolysis (blue) and oxidative phosphorylation (red) during myofibroblast differentiation induced by TGFβ or AngII. **e** Quantification of glycolysis 12 h post-TGFβ or post-AngII. Percent change vs. Ad-βgal vehicle. **f** Outline of glycolysis depicting the metabolites: glucose-6-phosphate (G-6-P), fructose-6-phosphate (F-6-P), fructose-1,6-bisphosphate (F-1,6-BP), fructose-2,6-bisphosphate (F-2,6-BP), dihydroxyacetone phosphate (DHAP), glycerol-3-phosphate (G-3-P), glyceraldehyde-3-phosphate (GA3P), 1,3-bisphosphoglyceric acid (1,3-BPG), 3-phosphoglyceric acid (3-PG), and the enzymes: phosphofructokinase 2/fructose bisphosphatase 2 (PFK2/FBP2), phosphofructokinase 1 (PFK1). veh ($n = 15$), TGFβ ($n = 10$), AngII ($n = 8$) **g–m** Absolute concentration of glycolytic intermediates in $Mcu^{-/-}$ (Ad-Cre) and control (Ad-βgal) MEFs at baseline and 12 h post-TGFβ. $n = 3$. **n, o** Adenoviruses co-expressing mutant PFK2/FBP2 and GFP: phosphatase-deficient PFK2/FBP2 (S32A, H258A; Ad-Glyco-High) or kinase-deficient PFK2/FBP2 (S32D, T55V; Ad-Glyco-Low). **p** $Mcu^{-/-}$ and control MEFs transduced with Ad-Glyco-High, Ad-Glyco-Low, or control Ad-GFP and 24 h later assayed for glycolysis by measuring ECAR; Ad-bgal + Ad-GFP ($n = 14$), Ad-bgal + Ad-Glyco-High ($n = 11$), Ad-bgal + Ad-Glyco-Low ($n = 9$), Ad-cre + Ad-GFP ($n = 14$), Ad-cre + Ad-Glyco-High ($n = 13$), Ad-cre + Ad-Glyco-Low ($n = 16$) **q, r** MEFs were transduced with Ad-Glyco-High and 24 h later treated with TGFβ or AngII for 24 h. Immunofluorescence was performed for α-SMA. Representative images are presented; white arrows denote α-SMA+/GFP+ cells, i.e., cells infected with Ad-Glyco-High that expressed α-SMA. Percentage of α-SMA +/GFP+ (Ad-Glyco-High) and α-SMA+/GFP− (Ad-Control) was quantified; $n = 3$ experiments, >50 cells per group. **s, t** MEFs were transduced with Ad-Glyco-Low and 24 h later treated with TGFβ or AngII for 24 h. Immunofluorescence was performed for α-SMA. Representative images are presented; white arrows denote α-SMA-/GFP+ cells, i.e., cells infected with Ad-Glyco-Low but did not express α-SMA. Percentage of α-SMA+/GFP+ (Ad-Glyco-Low) and α-SMA+/GFP− (Ad-Control) was quantified; $n = 3$ experiments, >50 cells per group. All data shown as mean ± SEM. ***$p < 0.001$, **$p < 0.01$, *$p < 0.05$ vs. vehicle control analyzed by ANOVA. ###$p < 0.001$, ##$p < 0.01$, #$p < 0.05$ vs. Ad-βgal analyzed by $t$-test. Scale bar = 50 μm. Also see Supplementary Fig. 3

formation[47]. Furthermore, these enhanced accessible regions contained binding sites for known pioneer transcription factors associated with fibroblast activation (example: Fig. 5e).

To determine the physiological relevance of αKG-dependent histone demethylation on myofibroblast differentiation we incubated MEFs in media containing cell-permeable dimethyl-αKG (DM-αKG) +/− TGFβ for 48 h and assessed α-SMA expression by immunofluorescence. Strikingly, DM-αKG increased the percentage of α-SMA positive cells to the same extent as 48 h of TGFβ treatment (Fig. 5h–j). Finally, we examined the effect of JIB-04, a cell-permeable inhibitor of JmjC-KDMs, on myofibroblast differentiation. JIB-04 is a selective inhibitor of specific JmjC-KDMs (Fig. 5k)[50], including JMJD3 (*KDM6B* gene), which demethylates H3K27me2/3[51]. $Mcu^{fl/fl}$ MEFs were transduced with Ad-Cre or Ad-βgal and 5 days later treated with TGFβ +/− 1 μM JIB-04 for 24 h and α-SMA expression was measured by immunofluorescence. Treatment with JIB-04 attenuated TGFβ-induced myofibroblast differentiation and inhibited the increased percentage of α-SMA positive cells observed in $Mcu^{-/-}$ MEFs (Fig. 5l–o). Altogether, these data demonstrate that TGFβ-induced metabolic changes lead to increased αKG levels and subsequent demethylation of repressive H3K27me2 chromatin marks to allow for coordinated genetic reprogramming and myofibroblast differentiation.

**Adult deletion of fibroblast *Mcu* worsens cardiac fibrosis after injury.** To directly examine myofibroblast differentiation in vivo, $Mcu^{fl/fl}$ mice were crossbred with a fibroblast-specific (*Col1a2* cis-acting fibroblast-specific enhancer with minimal promoter), tamoxifen (tamox)-inducible Cre transgenic mouse (Col1a2-CreERT) (Fig. 6a). The Col1a2-CreERT transgenic mice only expresses Cre in the fibroblast population in genetic fate mapping experiments[52]. Following tamoxifen administration, cardiac fibroblasts isolated from $Mcu^{fl/fl}$ × Col1a2-CreERT adult mice showed a near complete loss of MCU (Fig. 6b). CIII (subunit UQCRC2) was used as a mitochondrial loading control. We evaluated the role of cardiac fibroblast MCU using two in vivo models known to promote myofibroblast formation and cardiac fibrosis–myocardial infarction (MI) and chronic infusion of AngII.

MI results in significant cell death, initiating myofibroblast differentiation to generate a fibrotic scar to replace lost myocytes[6]. Mice were injected intraperitoneal (i.p.) with tamox

(40 mg/kg) for 10 days followed by a 10 day rest period before acquisition of baseline echocardiography. One week later mice underwent surgical ligation of the left coronary artery (LCA) to induce a large MI and left ventricular (LV) structure and function was tracked weekly by echocardiography (Fig. 6c). In both experimental and control groups, MI induced significant cardiac dysfunction and this was exacerbated in $Mcu^{fl/fl}$ × Col1a2-CreERT mice (Fig. 6d–f). Loss of fibroblast MCU ($Mcu^{fl/fl}$ × Col1a2-CreERT) significantly increased LV dilation, evident by increased LV end-diastolic diameter (LVEDD) and end-systolic diameter (LVESD), as well as reduced fractional shortening (FS) 2–4 weeks post-MI, as compared to Col1a2-CreERT controls (Fig. 6d–f). All other echocardiographic parameters are reported in Supplementary Fig. 8a–c and Supplementary Table 2. Loss of fibroblast MCU significantly increased heart weight to tibia length ratios (HW/TL) and lung edema (wet–dry lung weight) 4 weeks post-MI, suggesting an increase in hypertrophy and/or edema and inflammation, both of which are associated with fibrosis (Fig. 6g, h)[7]. Masson's trichrome staining of mid-ventricle cross-sections revealed increased collagen deposition in $Mcu^{fl/fl}$ × Col1a2-CreERT mice compared to Col1a2-CreERT controls (Fig. 6i). Quantification of fibrosis in the border and remote zones revealed a more than 2.5-fold increase in $Mcu^{fl/fl}$ × Col1a2-CreERT hearts vs. Col1a2-CreERT controls (Fig. 6j). Importantly, the increased fibrosis was associated with enhanced myofibroblast formation, which we assessed by immunofluorescence staining for α-SMA and CD31 (PECAM-1, marker of endothelial cells). Using this technique, blood vessels/smooth muscle cells costain with α-SMA and CD31, while myofibroblasts only stain with α-SMA (Supplementary Fig. 8d)[47]. $Mcu^{fl/fl}$ × Col1a2-CreERT hearts displayed increased myofibroblast number compared to Col1a2-CreERT controls in the remote zone 4 weeks post-MI (Fig. 6k).

To further define the centrality of $_mCa^{2+}$ exchange in myofibroblast formation we employed AngII-infusion as a second model of fibrosis. AngII is a direct stimulus of myofibroblast formation, and neurohormonal stress resulting from chronic increases in AngII levels is documented to induce cardiac fibrosis both clinically and experimentally[53]. Mice were injected i.p. with tamox (40 mg/kg) for 10 days followed by a 10 day rest period before subcutaneous implantation of Alzet mini-osmotic pumps to deliver AngII (1.1 mg/kg/day) for 4 weeks (Fig. 6l). Mice were sacrificed after 4 weeks and hearts were fixed and stained for fibrosis. Masson's trichrome staining of mid-ventricle

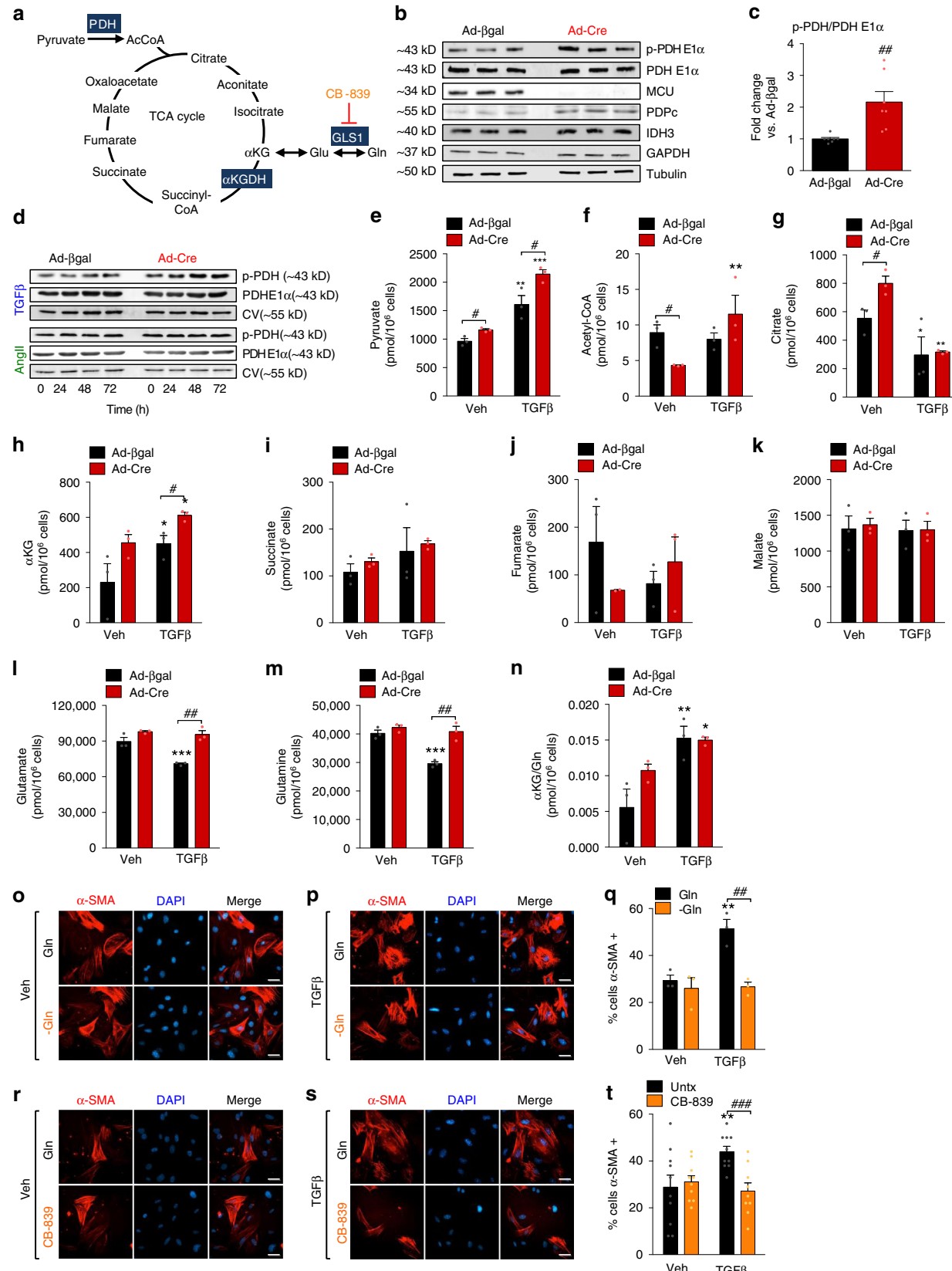

cross-sections revealed increased collagen deposition throughout the heart in $Mcu^{fl/fl} \times$ Col1a2-CreERT mice compared to Col1a2-CreERT controls (Fig. 6m). Quantification of interstitial fibrosis revealed a significant increase in $Mcu^{fl/fl} \times$ Col1a2-CreERT hearts vs. Col1a2-CreERT controls (Fig. 6n). In addition, chronic AngII significantly increased myofibroblast formation in $Mcu^{fl/fl} \times$ Col1a2-CreERT hearts vs. Col1a2-CreERT as determined by $\alpha$-SMA[+]/CD31[−] immunohistochemistry staining (Fig. 6o).

**Fig. 4** Loss of $_mCa^{2+}$ uptake reduces pyruvate entry into the TCA cycle. **a** TCA cycle with emphasis on key $_mCa^{2+}$-control points – pyruvate dehydrogenase (PDH) and α-ketoglutarate dehydrogenase (αKGDH). **b** $Mcu^{-/-}$ (Ad-Cre) and control (Ad-βgal) MEFs were immunoblotted for p-PDH E1α (phosphorylated pyruvate dehydrogenase, inactivate), total PDH E1α, PDPc (pyruvate dehydrogenase phosphatase catalytic subunit 1), IDH3A (mitochondrial isocitrate dehydrogenase subunit alpha), GAPDH (glyceraldehyde 3-phosphate dehydrogenase) and tubulin. **c** Ratio of p-PDH E1α/PDH E1α; $n = 7$. **d** $Mcu^{-/-}$ and control MEFs were treated with TGFβ or AngII for 0, 24, 48, or 72 h and immunoblotted for p-PDH E1α, PDH E1α and OxPhos Complex V. **e–n** Absolute concentration of metabolites in $Mcu^{-/-}$ (Ad-Cre) and control (Ad-βgal) MEFs at baseline and 12 h post-TGFβ; $n = 3$. **o–q** MEFs were cultured in media with or without Glutamine (Gln) and treated with TGFβ for 48 h. Immunofluorescence was performed for α-SMA. Representative images are presented. Percentage of α-SMA+ cells was quantified; $n = 3$ experiments, >50 cells quantified. **r–t** MEFs were cultured in media with or without CB-839, a potent and selective inhibitor of glutaminase 1 (GSL1) (see panel **a**). Immunofluorescence was performed for α-SMA. Representative images are presented. Percentage of α-SMA+ cells was quantified; $n = 10$ each >50 cells quantified. All data shown as mean ± SEM. ***$p < 0.001$, **$p < 0.01$, *$p < 0.05$ vs. vehicle control analyzed by ANOVA. ###$p < 0.001$, ##$p < 0.01$, #$p < 0.05$ vs. Ad-βgal (panels **c–n**), vs. –Gln (panel **q**), vs. CB-839 (panel **t**) analyzed by $t$-test. Scale bar = 50 μm

## Discussion

Recently, the $_mCa^{2+}$ field has been transformed by the discovery of many genes that encode $_mCa^{2+}$ transporters and channels. The biophysical properties of mtCU-mediated $Ca^{2+}$ influx have been extensively studied in many cell types, and the role of $_mCa^{2+}$ as a regulator of bioenergetics and cell death is well documented[16,17,26,54,55]. Here, we link changes in mtCU $Ca^{2+}$ uptake and mitochondrial metabolism with epigenetic modulation of the gene program to drive cellular differentiation. This study provides evidence that extracellular fibrotic signaling alters mitochondrial function in order to drive transcriptional changes in the nucleus necessary for differentiation.

Loss of $_mCa^{2+}$ uptake was sufficient to promote fibroblast to myofibroblast conversion and enhance the myofibroblast phenotype. Fibroblast-specific deletion of $Mcu$ in adult mice augmented myofibroblast formation and fibrosis post-MI and chronic AngII administration. Further, we found that fibrotic agonists signal to acutely downregulate $_mCa^{2+}$ uptake by rapidly increasing expression of the mtCU gatekeeper, MICU1. Although attributed to another mechanism, TGFβ-mediated reduction of $_mCa^{2+}$ uptake was also observed in smooth muscle cells–pretreatment with TGFβ reduced $_mCa^{2+}$ uptake in the face of increased $_cCa^{2+}$[56]. Given the noted role of MICU1 to negatively regulate uptake at signaling levels of $_cCa^{2+}$ [<2 μm], we hypothesize that fibrotic agonists signal to acutely inhibit $_mCa^{2+}$ uptake to initiate myofibroblast differentiation[26,28,30,57,58]. Our data suggest that extracellular stimuli are regulating cellular processes by directly altering mitochondrial signaling. We hypothesize that modulation of the uniporter is essential for the coordinated activation of both mitochondrial and cytosolic signaling pathways to mediate cellular differentiation. The outcome of this is two-fold. In addition to essential changes in mitochondrial metabolism upstream of epigenetic reprogramming, modulation of the mtCU is a way to enhance canonical cytosolic signaling pathways, hence the slight increase in NFAT activation (Supplementary Fig. 1).

Examination into mechanisms of pluripotency vs. differentiation has revealed the importance of metabolism, prompting us to evaluate the relationship between $_mCa^{2+}$ uptake, metabolism, and myofibroblast differentiation. Fibrotic agonists increased glycolysis and loss of MCU augmented this phenotype. Mechanistically, using mutant PFK2/FBP2 transgenes to increase or decrease glycolysis, we showed that enhanced glycolysis is sufficient to promote differentiation, whereas inhibition of glycolysis reverted the gain-of-function phenotype noted in $Mcu^{-/-}$ fibroblasts. This data is consistent with other studies that have shown glycolytic reprogramming correlates with myofibroblast differentiation[34,35]. Glycolytic reprogramming is a well-substantiated phenomenon which allows for diversion of glycolytic intermediates into ancillary metabolic pathways in order to generate building blocks for biosynthesis of macromolecules[59].

Our data suggest that increased glycolytic flux is necessary to fulfill cellular anabolic needs, for example nucleotide synthesis, de novo protein translation, membrane formation, etc., required for myofibroblast formation. We hypothesize that loss of $_mCa^{2+}$ uptake promoted aerobic glycolysis by reducing the activity of key $Ca^{2+}$-dependent enzymes. Indeed the phosphorylation status of PDH in response to fibrotic agonists and $Mcu^{-/-}$ fibroblasts suggested inactivity and thereby pyruvate was hindered from entering the TCA cycle. In correlation with our results, data obtained from ovarian cancer cell lines showed that MICU1 expression promoted the inhibition of PDH and aerobic glycolysis[60].

Metabolomic analysis revealed a multitude of changes induced by both TGFβ and the loss of MCU. In addition to increased levels of pyruvate, consistent with inactive PDH, metabolite quantification showed increased αKG ~2-fold in TGFβ-treated fibroblasts and this increase was augmented by loss of $_mCa^{2+}$ uptake. αKG is not restricted to its role as a TCA cycle intermediate, but is also a powerful signaling molecule. Of particular interest is the role of αKG in promoting DNA and histone demethylation by modulating αKG-dependent TET enzymes and JmjC-KDMs[44]. Previous studies have suggested that αKG regulates the balance between pluripotency and lineage-commitment of embryonic stem cells (ESCs). αKG maintained pluripotency of ESCs by promoting JmjC-KDM-dependent and TET-dependent demethylation, permitting gene expression to support pluripotency[18]. Interestingly, αKG also accelerated the differentiation of primed human pluripotent stem cells[20]. TGFβ and loss of MCU induced dynamic changes in histone lysine methylation at residues regulated by JmjC-KDMs. Furthermore, TGFβ increased chromatin accessibility at regions within key myofibroblast genes permitting increased gene transcription. TGFβ significantly reduced global H3K27me2 marks and $Mcu^{-/-}$ MEFs displayed reduced H3K27me2 compared to controls at baseline and post-TGFβ, suggesting these cells were primed for myofibroblast gene expression. Importantly, we determined that TGFβ induces the loss of H3K27me2 at regulatory myofibroblast gene loci (promoter and enhancer regions associated with gene activation and predicted binding sites for known fibrotic transcription factors). These data suggest that the observed increase in αKG bioavailability promotes H3K27me2 demethylation at myofibroblast-specific genes in order to promote differentiation. Further evidence in support of our working hypothesis are recent reports suggesting that JMJD3, a JmjC H3K27me2 demethylase with loci specificity, i.e., recruitment to lineage-specific genes, is dependent upon interaction with SMADs[61,62]. Since SMAD2/3 is a canonical transcription factor for TGFβ signaling and myofibroblast activation, it's intriguing to hypothesize that this interaction may provide myofibroblast-specific demethylation patterns in our model.

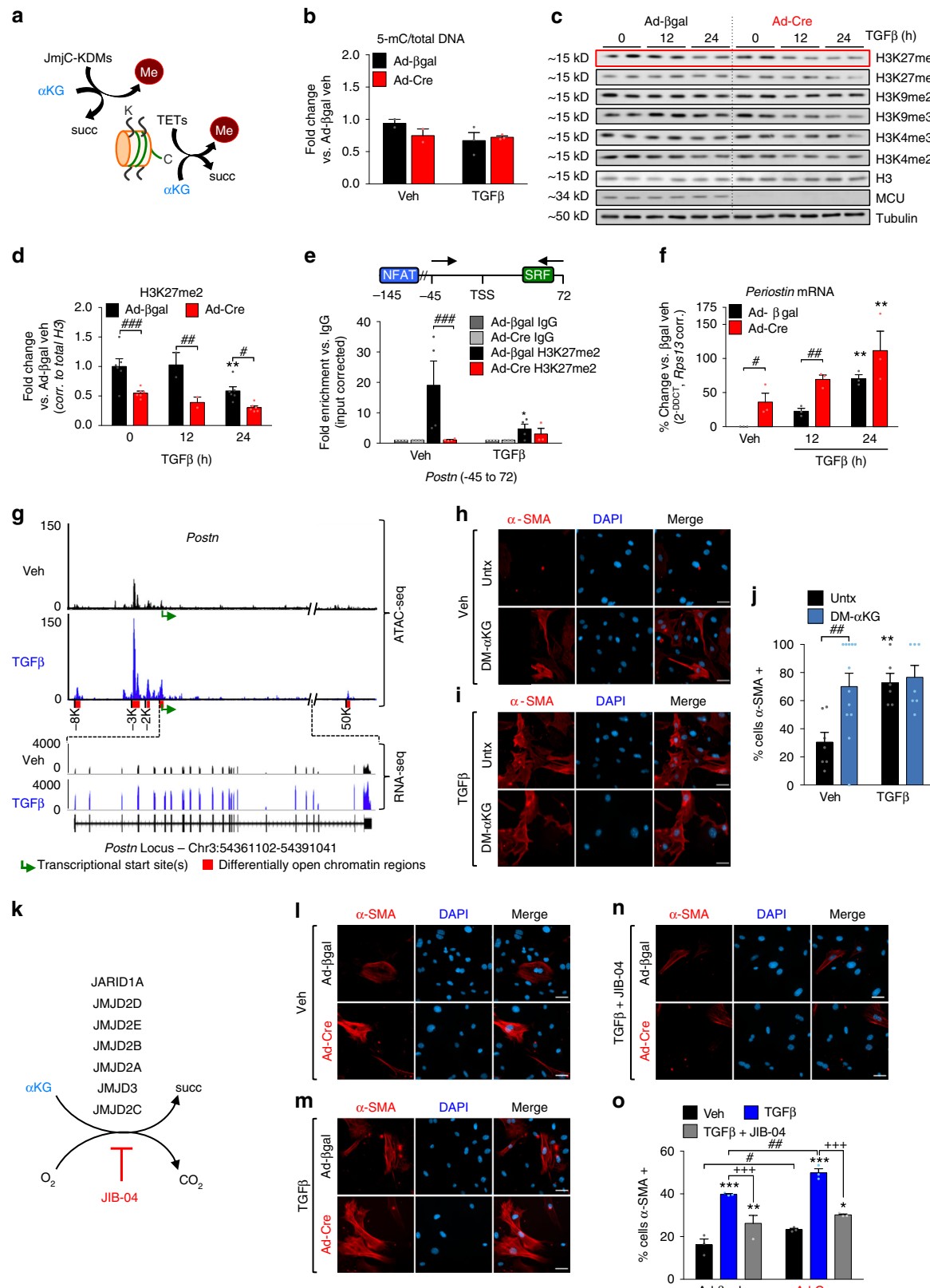

Since PDH-mediated pyruvate entry into the TCA cycle was inhibited, we suspect that anaplerotic pathways are being activated to replenish TCA cycle intermediates, specifically αKG. Our data suggest that the increased level of αKG associated with differentiation is likely generated through the pyruvate carboxylase pathway and/or glutaminolysis[59]. Pyruvate carboxylase activity is documented in cancer cells to mediate glucose-derived pyruvate to enter the TCA cycle at the level of oxaloacetate[63]. The second major replenishment pathway is through glutaminolysis which is a two-step process that converts glutamine to glutamate to αKG[41,42]. Our data suggests this is a more likely scenario, as we observed an increased αKG/Gln ratio post-TGFβ, and removal of

**Fig. 5** Loss of $_mCa^{2+}$ uptake drives myofibroblast differentiation through epigenetic reprogramming. **a** Simplified schematic of the reaction mechanism of α-ketoglutarate (αKG)-dependent dioxygenases: ten-eleven translocation (TET) enzymes and Jumonji-C (JmjC)-domain-containing demethylases (JmjC-KDMs). **b** Levels of 5-methylcytosine (5-mC) were measured in $Mcu^{-/-}$ (Ad-Cre) and control (Ad-βgal) MEFs by ELISA. Fold change vs. Ad-βgal veh. **c** MEFs were treated with TGFβ for 0, 12 or 24 h and cell lysates were immunoblotted for specific methylated histone 3 lysine (H3K) residues. Total H3 and Tubulin were used as loading controls. **d** Quantification of H3K27me2 protein expression. Band density was normalized to total H3. **e** H3K27me2 chromatin immunoprecipitation followed by qPCR (ChIP-qPCR) of *Periostin* in $Mcu^{-/-}$ (Ad-Cre) and control (Ad-βgal) MEFs at baseline (veh) and following 12 h TGFβ. Schematic shows loci of qPCR primers in relationship to myofibroblast-associated transcription factor binding sites – NFAT (nuclear factor of activated T-cells), SRF (serum response factor). **f** Expression of periostin (*Postn*) mRNA in $Mcu^{-/-}$ (Ad-Cre) and control (Ad-βgal) MEFs at baseline (veh) and post-TGFβ. **g** WT MEFs were treated with vehicle or TGFβ and assessed for chromatin accessibility and transcription using ATAC-seq and RNA-seq. Results of the *Postn* locus are shown. The height of the genome browser tracks shows the number of reads normalized by read depth and overall peak enrichment in the library. **h–j** Wildtype MEFs treated +/− cell-permeable, dimethyl-αKG and +/− TGFβ for 48 h followed by immunofluorescence for α-SMA. Representative images and quantification of percentage of α-SMA+ cells are shown. **k** Schematic of JmjC-KDM reactions indicating the specific JmjC-KDMs inhibited by JIB-04. **l–o** $Mcu^{-/-}$ (Ad-Cre) and control (Ad-βgal) MEFs were treated with vehicle, TGFβ, or TGFβ + 1 μM JIB-04 for 24 h and immunofluorescence was performed by costaining with α-smooth muscle actin (α-SMA) antibody (red) and DAPI (blue). Representative images and quantification of percentage of α-SMA+ cells are shown. $n = 3$ experiments for all quantified data. All data shown as mean ± SEM. ***$p < 0.001$, **$p < 0.01$, *$p < 0.05$ vs. vehicle control analyzed by ANOVA. ###$p < 0.001$, ##$p < 0.01$, #$p < 0.05$ vs. Ad-βgal analyzed by t-test. +++$p < 0.001$ TGFβ vs. TGFβ + JIB-04 analyzed by ANOVA. Scale bar = 50 μm

glutamine from culture media or pharmacological inhibition of glutaminase was sufficient to block myofibroblast formation.

In addition to providing carbons to the TCA cycle through αKG, glutamine metabolism contributes to many other cellular processes such as nucleotide synthesis, amino acid production, fatty acid synthesis, and redox modulation[64]; all cellular processes that are needed in the differentiating cell. Interestingly, in cancer cells increases in aerobic glycolytic flux is often associated with enhanced glutaminolysis[41,65]. Given the similarities with our model, it's intriguing to conjecture that the mtCU may play a similar role in these cell systems. The physiological relevance of the mtCU-dependent metabolic shift described here likely extends beyond epigenetic signaling pathways as wound healing and fibrosis typically occurs in a hypoxic environment and thus increased anaerobic glycolysis would be essential for energetic support in the differentiating fibroblast. However, our causative experiments downstream of metabolism indicate that this alone does not account for our phenotype.

In summary, we show that loss of $_mCa^{2+}$ uptake promotes myofibroblast differentiation both in vitro and in vivo. Until now, the role of $_mCa^{2+}$ uptake in cellular differentiation or epigenetic regulation has not been explored, but our study reveals its importance in myofibroblast differentiation through concerted alterations in both metabolism and epigenetics. In addition, our findings support an endogenous role for decreased mtCU-mediated $_mCa^{2+}$ uptake as an essential element of the differentiation process (Fig. 7). While much work remains to fully elucidate the role of mtCU during cellular differentiation, our current study provides a new framework underlying mitochondrial signaling and regulation of the epigenome.

## Methods
### Generation of fibroblast-specific Mcu conditional knockout mice.
Generation of $Mcu^{fl/fl}$ was previously reported[17]. $Mcu^{fl/fl}$ mice were crossed with fibroblast-specific Cre transgenic mice, Col1a2-CreERT, to generate tamoxifen-inducible, fibroblast-specific $Mcu$ knockouts. For temporal deletion of $Mcu$, mice 8–12 weeks of age were injected intraperitoneal with tamoxifen (40 mg/kg/day) for ten consecutive days. All mouse genotypes, including controls, received tamoxifen. All animal work complied with ethical regulations for animal testing and research, and was done in accordance with IACUC approval by Temple University and followed all AAALAC guidelines.

### Mouse embryonic fibroblast isolation and culture.
Mouse embryonic fibroblasts (MEFs) were isolated from $Mcu^{fl/fl}$ or C57/BL6 (WT) mice. Embryos were isolated from pregnant females at E13.5. The embryos were decapitated and all the red organs removed. Tissue was minced and digested in 0.25% trypsin supplemented with DNase for 15 min at 37 °C in the presence of 5% $CO_2$. Digested tissue was gently agitated by pipetting to dissociate cells. Cells from each embryo were suspended in Dulbecco's Modified Eagle's Medium (DMEM, Corning 10-013-CV) supplemented with 10% fetal bovine serum (FBS, Gemini Bio-Products), 1% penicillin/streptomycin (Sigma), and 1% Non-Essential Amino Acids (Gibco), plated on a 10 cm dish and incubated at 37 °C in the presence of 5% $CO_2$. In some experiments MEFs were incubated in media containing dimethyl-αKG (dimethyl 2-oxoglutarate, Sigma, 349631), JIB-04 (Sigma, SML0808), or CB-839 (Cayman, 1439399-58-2) as indicated in the results and figures. For all imaging studies, MEFs were plated on collagen-coated 35 mm dishes (Mattek).

### Cardiac fibroblast isolation and culture.
After euthanasia, hearts were excised from mice and rinsed with cold Hank's Balanced Salt Solutions (HBSS, Corning, MT21023CV). Atria were removed and ventricles were placed into HBSS containing 150 units/ml of Collagenase Type 2 (Worthington, LS004176) and 0.6 mg/ml Trypsin (USB Corp., 22705). The ventricles were minced into small pieces to facilitate digestion, transferred to a small beaker and incubated in a shaking 37 °C water bath for 5 min. The supernatant was aspirated and discarded and new HBSS/enzyme solution was added to the beaker. Beaker was incubated in a shaking 37 °C water bath for 15 min and supernatant was collected and transferred to a conical containing media and FBS. This was repeated at least 3 more times until remaining pieces were too small to separate from digestion solution. Cells were spun down at 400×g for 5 min. Supernatant was discarded, pellets were resuspended in FBS, and spun down at 400×g for 5 min. After centrifugation, supernatant was discarded and pellets were resuspended in DMEM (Corning 10-013-CV) supplemented with 10% fetal bovine serum (FBS, Gemini Bio-Products), 1% penicillin/streptomycin (Sigma), and 1% Non-Essential Amino Acids (Gibco). Resuspended cells were passed through a 70-micron cell strainer, placed into a cell culture dish, and incubated at 37 °C in the presence of 5% $CO_2$. After ~1 h, media was removed and replaced with fresh media.

### Adenoviral transfer.
For experiments that required adenoviral gene transfer, MEFs were incubated in adenovirus for 24 h at which time the media was changed. To knockout $Mcu$, MEFs were transduced with adenovirus encoding Cre-recombinase (Ad-Cre) or βgalactosidase (Ad-βgal) for 24 h and experiments were performed 5 days post-infection in order to ensure sufficient time for protein turnover. For experiments using adenovirus encoding Glyco-High, Glyco-Low, or mito-R-GECO1, cells were incubated for an additional 24 h prior to the experiment.

The following adenoviruses have previously been described: NFAT-c1-GFP, Glyco-High, Glyco-Low, mito-R-GECO1[38,39,66,67]. Glyco-High and Glyco-Low adenoviruses were made and purified by Vector Labs (Malvern, PA) using cDNA for a rat liver PFKFB1 isoform of phosphofructokinase 2 (PFK2)/fructose-2,6-bisphosphatase (FBP2). The Glyco-High adenovirus has 2 single-amino acid point mutations (S32A and H258A) which result in the enzyme having only PFK2 activity, while the Glyco-Low adenovirus has 2 single-amino acid point mutations (S32D and T55V) which result in the enzyme having only FBP2 activity[38,39]. Ad-GFP was purchased from Vector Labs (Malvern, PA).

### Myofibroblast differentiation.
Myofibroblast differentiation was induced using 10 ng ml$^{-1}$ recombinant mouse transforming growth factor-β (TGFβ, R&D Systems, 7666-MB-005) or 10 μM Angiotensin II (AngII, Sigma A9525). In all experiments, FBS was reduced to 1% 24 h prior to and during treatment with TGFβ or AngII.

### Western blot analysis.
All protein samples were lysed by homogenization in RIPA buffer supplemented with phosphatase inhibitors (Roche, 4906837001) and

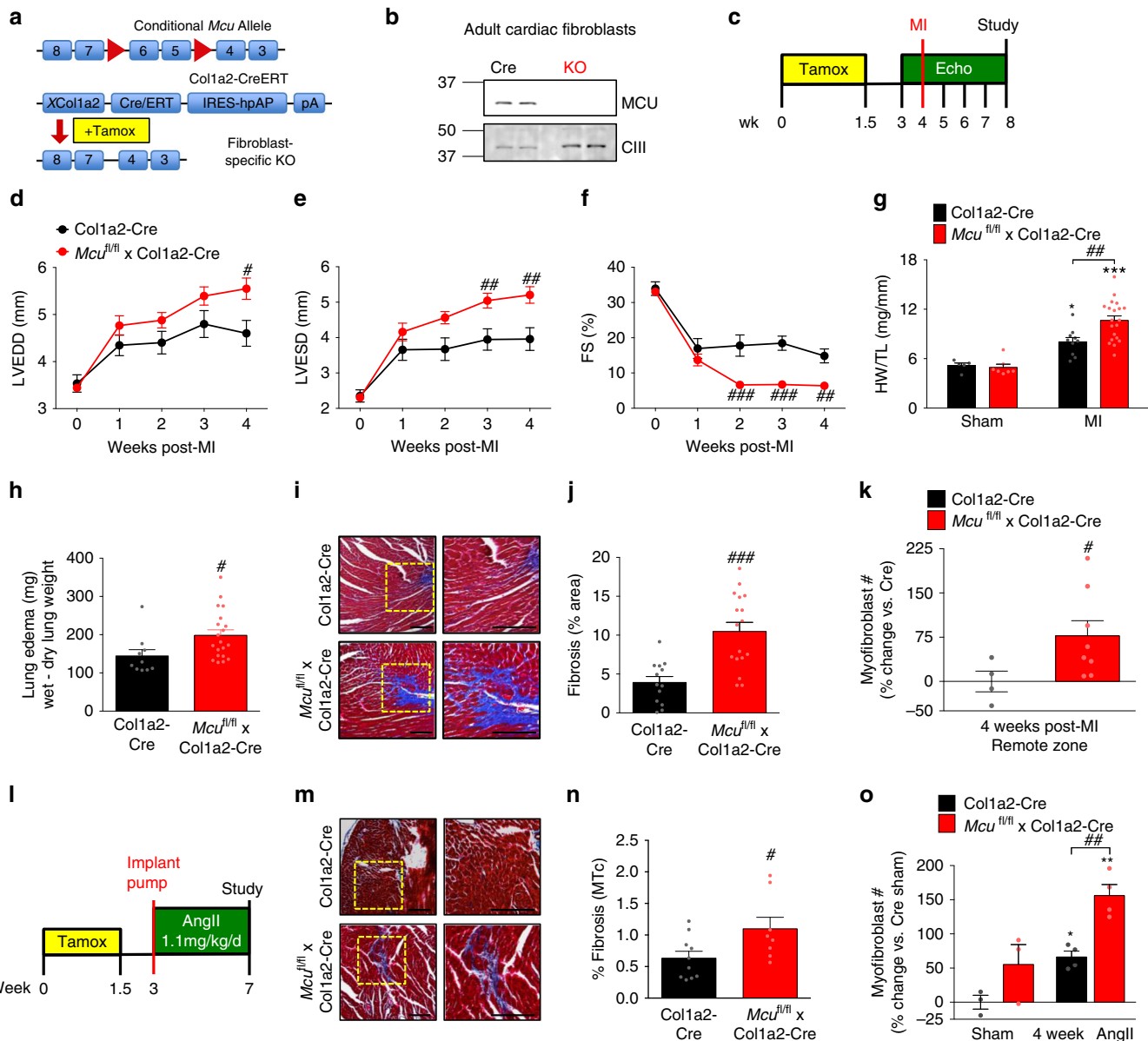

**Fig. 6** Adult deletion of fibroblast *Mcu* exacerbates cardiac dysfunction, fibrosis, and myofibroblast formation post-MI and chronic angiotensin II administration. **a** *Mcu*^fl/fl mice were crossed with mice expressing a tamoxifen (tamox)-inducible, fibroblast-specific Cre recombinase (Col1a2-CreERT). Tamox administration (40 mg/kg/day) for 10 days induces fibroblast-restricted Cre expression. **b** Adult cardiac fibroblasts were isolated post-tamox treatment and immunoblotted for MCU expression. CIII (Complex III, subunit-UQCRC2) was used as a loading control. **c** Experimental timeline: 8–12-week-old mice were treated with tamox and allowed to rest before permanent ligation of the left coronary artery. **d–f** Cardiac function was analyzed by echocardiography 1 week prior to MI and every week thereafter. M-mode echo measurements of left ventricular end diastolic diameter (LVEDD), left ventricular end systolic diameter (LVESD), and percent fractional shortening (FS). $n = 10$ Col1a2-Cre, $n = 20$ *Mcu*^fl/fl × Col1a2-Cre. **g** Ratio of heart weight to tibia length 4 weeks post-MI. Sham: $n = 5$ Col1a2-Cre, $n = 7$ *Mcu*^fl/fl × Col1a2-Cre; post-MI: $n = 10$ Col1a2-Cre, $n = 20$ *Mcu*^fl/fl × Col1a2-Cre. **h** Quantification of wet–dry lung weight as a measurement of lung edema 4 weeks post-MI. $n = 10$ Col1a2-Cre, $n = 20$ *Mcu*^fl/fl × Col1a2-Cre. **i, j** LV sections were stained with Masson's trichrome (MTc). Representative images are shown. Percent fibrotic area per infarct border and remote zones. $n = 4$ mice per group, multiple non-consecutive heart sections were quantified per mouse. **k** Percent change in myofibroblast number (α-SMA+/CD31−) in the remote zone 4 weeks post-MI. $n = 4$ Col1a2-Cre, $n = 8$ *Mcu*^fl/fl × Col1a2-Cre; multiple non-consecutive heart sections in the remote zone were quantified per mouse. **l** Experimental timeline: mini-osmotic pumps were subcutaneously implanted in mice to deliver AngII for 4 weeks. **m, n** Representative images of MTc stained LV sections. Percent fibrosis per area was quantified. $n = 5$ Col1a2-Cre, $n = 4$ *Mcu*^fl/fl × Col1a2-Cre; multiple non-consecutive heart sections were quantified per mouse. **o** Percent change in myofibroblast number (α-SMA+/CD31−) 4 weeks post-AngII infusion. Sham: $n = 3$; AngII: $n = 4$; multiple non-consecutive heart sections were quantified per mouse. All data shown as mean ± SEM, ***$p < 0.001$, **$p < 0.01$, *$p < 0.05$ vs. control (week 0 or sham) analyzed by ANOVA. ###$p < 0.001$, ##$p < 0.01$, #$p < 0.05$ vs. Ad-βgal analyzed by *t*-test. Scale bar = 250 μm. Also see Supplementary Fig. 7 and Supplementary Table 2

protease inhibitors (Sigma, S8830). Samples were sonicated briefly and centrifuged at 5000×*g* for 10 min. The supernatant was collected and used for further analysis. Protein amount was quantified using the Bradford Protein Assay (Bio-Rad) and equal amounts of protein (10–50 μg) were run by electrophoresis on poly-acrylamide Tris-glycine SDS gels. Gels were transferred to PVDF (EMD Millipore,

IPFL00010) and membranes were blocked for 1 h in Blocking Buffer (Rockland, MB-070) followed by incubation with primary antibody overnight at 4 °C. Membranes were washed in TBS-T 3 times for 5 min each and then incubated with secondary antibody for 1 h at room temperature. After incubation with fluorescent secondary antibodies, membranes were washed in TBS-T 3 times for 5 min each

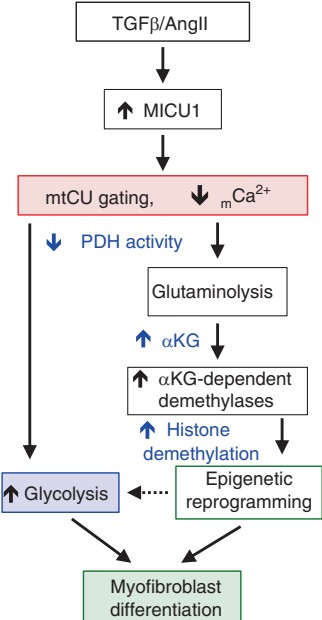

**Fig. 7** Enhanced mtCU gating is essential for myofibroblast differentiation. Signaling model for myofibroblast differentiation whereby fibrotic stimuli acutely upregulates MICU1 to limit $_mCa^{2+}$ uptake. Increased MICU1-dependent mtCU gating leads to a cascade of metabolic changes necessary for myofibroblast differentiation. Decreased [$_mCa^{2+}$] downregulates the activity of $Ca^{2+}$-dependent dehydrogenases (PDH, αKGDH). This elicits an increase in glycolysis, which supports energetic demands during the differentiation process and the activation of ancillary biosynthetic pathways to support conversion into a myofibroblast. In addition, there are distinct changes in the levels of TCA cycle intermediates, including increased αKG bioavailability, which drives JmjC-KDM-dependent histone demethylation for chromatin remodeling and activation of the myofibroblast gene program

and then imaged on a Licor Odyssey system. The following antibodies were used in the study: MCU (1:1000, Sigma-Aldrich, HPA016480), MCUB (1:250, Santa Cruz, sc-163985), MICU1 (1:500, Custom generation by Yenzyme, courtesy of the Madesh lab), MCUR1 (1:500, Custom generation, courtesy of the Madesh lab), EMRE (1:250, Santa Cruz, sc-86337), VDAC (1:1000, Abcam, ab15895), PDHE1α phospho S293 (1:1000, Abcam, ab110330), PDHE1α (1:1000, Abcam, ab110330), IDH3A (1:500, Abcam, ab58641), α-tubulin (1:1000, Abcam, ab7291), ETC respiratory chain complexes (1:2,500, OxPhos Cocktail, Abcam, MS604), H3K4me3 (1:2000, Millipore, 07-473), H3K9me3 (1:2000, Abcam, ab8898), H3K27me3 (1:2000, Cell Signaling, 9733), H3K4me2 (1:2000, Cell Signaling, 9726), H3K9me2 (1:2000, Cell Signaling, 4658), H3K27me2 (1:2000, Cell Signaling, 9728), H3 (1:2000, Cell Signaling, 4499); and Licor IRDye secondary antibodies: anti-mouse (1:12,000, 926–32210), anti-rabbit, (1:12,000, 926–68073), anti-goat (1:12,000, 926–32214). All full-length Western blots are displayed in Supplementary Figs. 9–13.

**Live cell imaging of $Ca^{2+}$ transients**. $Mcu^{fl/fl}$ MEFs were infected with Ad-Cre or Ad-βgal for 72 h and then transduced with adenovirus encoding a mitochondrial-targeted $Ca^{2+}$ reporter (Mito-R-GECO). Forty-eight hours post-infection with Mito-R-GECO, prior to live-cell imaging, MEFs were loaded with the calcium sensitive dye Fluo-4 AM (1 μM, Invitrogen) or Fura-2 (1 μM, Invitrogen) to measure cytosolic calcium transients. Cells were placed in a 37 °C heated chamber in physiological Tyrode's buffer (150 mM NaCl, 5.4 mM KCl, 5 mM HEPES, 10 mM glucose, 2 mM CaCl$_2$, 2 mM sodium pyruvate, pH 7.4) and imaged on a Carl Zeiss Axio Observer Z1 microscope. $Ca^{2+}$ transients were continuously recorded and analyzed on Zen software. After 2–3 min of baseline recording, a single pulse of 1 mM ATP was delivered to liberate intracellular $Ca^{2+}$ ($_iCa^{2+}$) stores. Background fluorescence was subtracted from each experiment before calculating the peak intensity as the maximal fluorescence/baseline fluorescence.

**Immunofluorescence**. MEFs were seeded on coated 35-mm dishes. MEFs were fixed for 15 min in 4% paraformaldehyde, then permeabilized for 15 min with 0.15% Triton-X-100, and blocked in PBS containing 10% goat serum for 1 h at room temperature. MEFs were incubated in primary antibody α-SMA (1:1000, Sigma-Aldrich, A2547) overnight at 4 °C and secondary antibody goat anti-mouse

Alexa Fluor 594 (1:1000, ThermoFisher, A-11005) for 45 min at 37 °C. Prior to imaging, MEFs were incubated with Hoechst 33342 to demarcate cell nuclei. Cells were imaged on a Carl Zeiss Axio Observer Z1 fluorescent microscope. Images were acquired in the red (590ex/617em) and blue (350ex/461em) channels. α-SMA expression was assessed by quantifying fluorescence intensity and the percentage α-SMA positive cells. More than 50 cells per dish were analyzed.

**Gel contraction**. Fibroblast contractile activity was assessed by collagen contraction assays in which 112,500 MEFs were seeded into a 2 mg/ml collagen type I (Corning, 354249) gel matrix and cast into a 48-well plate. Once collagen polymerized, the gel was gently released from edges of the well and media was added to the well. Images were taken using a Nikon SMZ1500 stereomicroscope at 0 and 24 after the gel was released from well edges. ImageJ software (NIH) was used to calculate the surface area, which is presented as percent gel contraction relative to initial size of the gel.

**Cell proliferation**. MEFs were seeded at the same density in 96-well plates and quantified using the CyQUANT NF Cell Proliferation Assay Kit (ThermoFisher) according to the manufacturer's protocol.

**qPCR mRNA analysis**. RNA was isolated using the RNeasy Mini Kit (Qiagen, 74104) according to the manufacturer's protocol. RNA (2 μg) was reverse transcribed into cDNA using the High Capacity cDNA Reverse Transcription Kit (ThermoFischer, 4368814) according to the manufacturer's protocol. Thermocycler conditions were as follows: 25 °C for 10 min, 37 °C for 2 h, 85 °C for 5 min. Quantification of cDNA was done using Luminaris HiGreen qPCR Master Mix (ThermoFischer, K0991) following the manufacturer's protocol. Cycling conditions were as follows: 95 °C for 10 min followed by 40 cycles of amplification (95 °C denaturation for 15 seconds, 60 °C annealing/extension for 1 min). We evaluated samples for mRNA expression of Collagen type I alpha 1 chain (*Col1a1*), Collagen type I alpha 2 chain (*Col1a2*), Collagen type III alpha 1 chain (*Col3a1*), α-SMA (*Acta2*), periostin (*Postn*), lysyl oxidase (*Lox*), fibronectin (*Fn1*), and platelet derived growth factor receptor alpha (*Pdgfra*). *Rps13* (Ribosomal Protein S13) was used as a housekeeping gene. All samples were analyzed in duplicate and averaged. Fold change in mRNA expression was measured using the Comparative $C_T$ Method (2^-ΔΔ$C_T$). Primers used are listed below in Table 1.

**NFAT translocation assay**. MEFs were plated on coated 35 mm dishes and infected with Ad-NFATc1-GFP for 24 h at which time live-cell images were taken followed by treatment with 10 ng ml$^{-1}$ TGFβ or 10 μM AngII for 24 h. For live-cell imaging, cells were placed in a 37 °C heated chamber on a Carl Zeiss Axio Observer Z1 fluorescent microscope. Prior to imaging, MEFs were incubated with Hoechst 33342 to demarcate cell nuclei. Images were acquired in the green channel (490ex/525em) and blue channel (350ex/460em). NFAT localization was quantified as the nuclear/cytoplasmic ratio of GFP fluorescence. More than 50 cells per dish were analyzed.

**Evaluation of $_mCa^{2+}$ uptake and efflux**. Before permeabilization, MEFs were washed in extracellular-like $Ca^{2+}$-free buffer (120 mM NaCl, 5 mM KCl, 1 mM KH$_2$PO$_4$, 0.2 mM MgCl$_2$, 0.1 mM EGTA, 20 mM HEPES-NaOH, pH 7.4). MEFs (1.5 million) were then transferred to intracellular-like medium (ICM) (120 mM KCl, 10 mM NaCl, 1 mM KH$_2$PO$_4$, 20 mM HEPES-Tris, protease inhibitors (Sigma EGTA-Free Cocktail), 5 mM succinate, 2 μM thapsigargin, 40 μg ml$^{-1}$ digitonin, 10 μM SB37157 (NCLX inhibitor), pH 7.2). ICM was cleared with Chelex 100 to remove trace $Ca^{2+}$ (Sigma). MEFs were gently stirred and 1 μM Fura-2 (ThermoFisher, F1200) was added to monitor extra-mitochondrial $Ca^{2+}$. At 20 seconds, JC-1 (Enzo Life Sciences) was added to monitor Δψ. Fluorescence signals were monitored in a temperature controlled (37 °C) multi-wavelength-excitation/dual-wavelength-emission spectrofluorometer (Delta RAM, Photon Technology Int.) using 490-nm excitation (ex)/535-nm emission (em) for the JC-1 monomer, 570-nm ex/595-nm em for the J-aggregate of JC-1, and 340-nm and 380-nm ex/510-nm em for Fura-2. At 350 seconds a $Ca^{2+}$ bolus was added and clearance of extra-mitochondrial $Ca^{2+}$ was representative of $_mCa^{2+}$ uptake. At completion of the experiment 10 μM of the protonophore FCCP was added to uncouple the Δψ and release matrix free-$Ca^{2+}$.

To quantify actual $Ca^{2+}$ content, a standard curve of $Ca^{2+}$ binding Fura-2 was generated from serial diluted $Ca^{2+}$ standards (0.01–120 μM) in ICM. The K$_d$ was calculated from the standard curve using GraphPad Prism 6.0 software. Fura-2 fluorescence ratio was converted to to [$Ca^{2+}$] by the following equation: [$Ca^{2+}$] = $K_d \times (R - R_{min})/(R_{max} - R) \times$ Sf2/Sb2. ($R_{min}$ (ratio in 0–$Ca^{2+}$) = 1.341; $R_{max}$ (ratio at saturation) = 27.915; Sf2 (380/510 reading in 0-$Ca^{2+}$) = 15822.14; Sb2 (380/510 reading with $Ca^{2+}$ saturation) = 1794.32). The percentage of initial $_mCa^{2+}$ uptake (200 s after $Ca^{2+}$ addition) was plotted against the bath $Ca^{2+}$ concentration for each of the different $Ca^{2+}$ boluses to generate a dose response curve.

**ECAR and OCR measurements**. A Seahorse Bioscience XF96 extracellular flux analyzer was employed to measure extracellular acidification rates (ECAR) and oxygen consumption rates (OCR). ECAR was measured using the Glycolytic Stress

### Table 1 qPCR Primers

| Gene | Forward primer 5′–3′ | Reverse primer 5′–3′ |
|---|---|---|
| *Rps13* | gcaccttgagaggaacagaa | gagcacccgcttagtcttatag |
| *Col1a1* | ttcagggaatgcctggtgaa | acctttgggaccagcatca |
| *Col1a2* | gaaaagggtccctctggagaa | aataccgggagcaccaagaa |
| *Col3a1* | tgctggaaagaatggggagac | ggtccagaatctcccttgtcac |
| *Acta2* | gtgaagaggaagacagcacag | gcccattccaaccattactcc |
| *Postn* | ccattggaggcaaacaactcc | ttgcttcctctcaccatgca |
| *Lox* | acgtcctgtgactatgggtac | tctgccgcataggtgtcata |
| *Fn1* | cgtcattgccctgaagaaca | aagggtaaccagttggggaa |
| *Pdgfra* | caaagggaggacgttcaagac | tgcgtccatctccagattca |
| *Micu1* | aagaacactccctgccattt | gccagggtcatctgcattat |
| *Mcu* | gatgacgtgacggtggttta | gtcagagataggcttgagtgtg |

Test Kit (Seahorse Biosciences) and OCR was measured using the Mito Stress Test Kit following the manufacturer's protocol. To evaluate ECAR, 20,000 MEFs/well were plated in XF media pH 7.4 without supplementation. Non-glycolytic acidification was measured, then 10 mM glucose was injected to measure basal glycolysis, followed by 3 μM oligomycin to inhibit mitochondrial ATP production and reveal maximal glycolytic capacity, and finally 50 mM 2-deoxy-glucose was injected to completely inhibit all glycolysis. To evaluate OCR, 20,000 MEFs/well were plated in XF media pH 7.4 supplemented with 10 mM glucose and 1 mM sodium pyruvate. Basal OCR was measured, then 3 μM oligomycin was injected to inhibit ATP-linked respiration, followed by 2 μM FCCP to measure maximal respiration, and finally 1.5 μM rotenone/antimycin A was injected to completely inhibit all mitochondrial respiration. After each experiment, protein concentration was measured and wells were normalized using the Wave software.

**Metabolomic profiling.** Cells in a 10 cm dish were washed with 5% (w/w) mannitol (10 ml for the first wash, 2 ml for the second wash) and extracted in 800 μl methanol plus 550 μl internal standard solution (Human Metabolome Technologies, HMT). Extracted solution was spun down at 2300 × g at 4 °C for 5 min. The supernatant was transferred into centrifugal filter units (HMT) and centrifuged at 9100 × g at 4 °C for ~3.5 h until no liquid remained in the filter cup. Filtrate was frozen at −80 °C and shipped to HMT for analysis by CE-TOFMS and CE-QqQMS (Boston, MA). Filtrate was centrifugally concentrated and resuspended in 50 μl of ultrapure water immediately before the measurement.

Cationic metabolites were analyzed using an Agilent CE-TOFMS system (Agilent Technologies) Machine No. 3 and a fused silica capillary (i.d. 50 μm × 80 cm) with Cation Buffer Solution (HMT) as the electrolyte. The sample was injected at a pressure of 50 mbar for 10 s. The applied voltage was set at 27 kV. Electrospray ionization-mass spectrometry (ESI-MS) was conducted in the positive ion mode, and the capillary voltage was set at 4000 V. The spectrometer was scanned from *m/z* 50 to 1,000.

Anionic metabolites were analyzed using an Agilent Capillary Electrophoresis System equipped with an Agilent 6460 TripleQuad LC/MS Machine No. QqQ3 and a fused silica capillary (i.d. 50 μm × 80 cm) with Anion Buffer Solution (HMT) as the electrolyte. The sample was injected at a pressure of 50 mbar for 25 s. The applied voltage was set at 30 kV. ESI-MS was conducted in the positive and negative ion mode, and the capillary voltage was set at 4000 V for positive and 3500 V for negative mode.

Peaks detected in CE-TOFMs analysis were extracted using automatic integration software (MasterHands ver.2.17.1.11 developed at Keio University) and those in CE-QqQMS analysis were extracted using automatic integration software (MassHunter Quantitative Analysis B.06.00 Agilent Technologies, Santa Clara, CA, USA) in order to obtain peak information including *m/z*, migration time, and peak area. The peak area was then converted to relative peak area by the following equation: Relative peak area = Metabolite Peak Area/(Internal Standard Peak Area × Normalization Factor). The peaks were annotated based on the migration times in CE and *m/z* values determined by TOFMS. Putative metabolites were then assigned from HMT metabolite database on the basis of *m/z* and migration time. All metabolite concentrations were calculated by normalizing the peak area of each metabolite with respect to the area of the internal standard and by using standard curves, which were obtained by three-point calibrations. A heat map was generated using ClustVis[68]. Unit variance was applied to rows. Rows were clustered using Manhattan distance and average linkage.

**NAD(P) and NAD(P)H assays.** NAD(P) and NAD(P)H ratios were measured using the bioluminescent NAD(P)/NAD(P)H-Glo Assay (Promega), performed according to the manufacturer's protocol. Briefly, cells were seeded and treated in a 96-well plate. Cells were lysed and split into separate wells to measure NAD(P) and NAD(P)H by selectively destroying the oxidized forms by heating in basic solution and the reduced forms in acidic solution. Utilizing the NAD cycling enzyme and reductase enzyme, the generated luciferin is used by the recombinant luciferase to produce light. The determine the redox state of the cell, the NAD(P):NAD(P)H ratio is reported.

**DNA methylation.** To extract genomic DNA, cells were collected and washed with PBS followed by 2 h incubation at 60 °C in DNA isolation buffer (0.5% SDS, 100 mM NaCl, 50 mM Tris pH 8, 3 mM EDTA, 0.1 mg/ml proteinase K). DNA was extracted using chloroform followed by ethanol precipitation and dissolved in double-distilled water. DNA methylation was quantified using the MethylFlash™ Methylated DNA Quantification Kit (Colorimetric), according to the manufacturer's protocol (Epigentek Inc.). One hundred nanograms of input DNA were used per reaction. Absorbance at 450-nm was measured using a Tecan Infinite F50 microplate reader.

**ChIP-qPCR.** ChIP-qPCR was performed using the ChIP-IT High Sensitivity Kit (Active Motif, 53040) according to the manufacturer's protocol. Cells were fixed, lysed and sonicated for 30 m (30 s on, 30 s off) leading to chromatin fragments between 200 and 1200 base pairs. DNA-bound protein was immunoprecipitated using 2 μg anti-H3K27me2 (Active Motif, clone MABI 0324) or IgG (Santa Cruz, 2025). Following immuneprecipitation, cross-links were reversed, protein was removed, and DNA was purified. qPCR was performed with equal amounts of H3K27me2-immunoprecipitated sample, IgG-immunoprecipitated sample, and input sample. Values were normalized to input measurements and fold enrichment was calculated. qPCR primers were designed to target gene loci regions flanking or nearby myofibroblast transcription factor predicted binding sites according to Genomatrix-MatInspector Software analysis. The following ChIP-qPCR primers were used: *periostin* forward primer 5′-CCACAGCCCAGAGCTATATAAAC-3′, *periostin* reverse primer 5′-CAGCAGCAGCAGAGCATATAA-3′, *platelet-derived growth receptor alpha* forward primer 5′-AGCAACTACACGGCACTTT-3′, *platelet-derived growth receptor alpha* reverse primer 5′-CTGGGCCTCGCTAGA AATATG-3′.

**RNA-seq/ATAC-seq.** RNA-seq and ATAC-seq were performed in cardiac fibroblasts (CFs) treated with or without TGFβ for 24 h; 3 biological replicates. For RNA-seq, total RNA was isolated from CFs using a standard RNA isolation kit (Qiagen). The TrueSeq stranded mRNA library kit was used to enrich polyA mRNAs via poly-T based RNA purification beads which were then amplified using Hiseq rapid SR cluster kit and multiplexed and run using the HiSeq rapid SBS kit. Reading depth was ~40 M reads per sample and single-end 75 bp fragments were generated for bioinformatic analysis using DESeq2 and assessed for quality control. For ATAC-seq, gDNA was isolated from the same treated samples and incubated with Tn5 transposomes which fragments and adds adapters simultaneously, in open chromatin regions. Deep sequencing of these purified regions provides 50 bp fragments and downstream base-pair resolution of nucleosome-free regions in the genome. ATAC-seq data was then processed (trimmed, filtered, and quality controlled) using the Illumina BaseSpace sequencing HUB and enriched regions were identified using MACS2 analysis. Only those enriched regions found across all 3 biological samples were included in the analysis. All kits were obtained from Illumina and all sequencing was performed on the Illumina HiSeq2500 sequencer. Both sequencing data sets were aligned to the mouse genome (mm10). For data visualization, BIGWIG files were generated for RNA-seq and ATAC-seq viewing in the Integrative Genomics Viewer (Version 2.5).

**Echocardiography.** Transthoracic echocardiography of the left ventricle was performed and analyzed on a Vevo 2100 imaging system (VisualSonics). Mice were anesthetized with 2% isoflurane in 100% oxygen during acquisition. M-mode images were collected in short-axis and analysis was performed using VisualSonics software.

**Myocardial infarction.** Ligation of the left coronary artery (LCA) was performed as described previously in Gao et al.[69]. Briefly, mice were anesthetized with 2% isoflurane in 100% oxygen and the heart exposed via a left thoracotomy at the fifth intercostal space. The LCA was permanently ligated to induce a large myocardial infarction and the heart was returned to the chest cavity.

**Chronic angiotensin II infusion.** Mice were anesthetized with 2% isoflurane in 100% oxygen and mini-osmotic pumps (Alzet Model 1004) were inserted subcutaneously to deliver 1.1 mg/kg/d Angiontesin II (Sigma, A9525) for 4 weeks.

**Tissue gravimetrics.** Mice were sacrificed followed by isolation and weighing of the heart and lungs, as well as measurement of tibia length. Heart gravimetrics were assessed by heart weight/tibia length ratios. Lungs were weighed at the time of isolation (wet lung weight) and after dehydration at 37 °C for 1 week (dry lung weight). Lung edema was quantified by subtracting wet–dry lung weight.

**Histology.** For histological analysis, hearts were collected at the indicated time points and fixed in 10% buffered formalin. Next, hearts were dehydrated and embedded in paraffin followed by collection of serial 7 μm sections. To evaluate

fibrosis, sections were stained with Masson's trichrome (Sigma). Sections were examined using a Nikon Eclipse Ni microscope and images were acquired with a high-resolution digital camera (Nikon DS-Ri1). The percentage of fibrosis was quantified using ImageJ software (NIH). Blue pixels were expressed as a percentage of the entire image surface area.

To quantify myofibroblasts, antigen retrieval was performed and sections were subsequently stained with anti-α-SMA antibody (1:1000, Sigma-Aldrich, A2547) and anti-CD31 (1:30, R&D Systems, AF3628). Sections were incubated with antibodies in a humidified chamber overnight at 4 °C followed by 1 h at room temperature. Sections were washed three times for 5 min each in PBS and incubated in secondary antibodies for 1 h at 37 °C in a humidified chamber. Secondary antibodies used were: Alexa Fluor 488 (1:250, Invitrogen, A21202) and Alexa Fluor 555 (1:100, Invitrogen, A21432). After washing three times for 5 min each, sections were stained with DAPI (Invitrogen R37606). After DAPI staining, sections were washed three times for 5 min and then incubated with Sudan black B (Abcam, ab146284) for 40 min at room temperature followed by 6 washes for 10 min each. Finally sections were mounted on slides using Vectashield. Images were taken using a Carl Zeiss Axio Observer Z1 fluorescent microscope. Images were acquired in the green channel (490ex/525em), orange channel (555ex/580em), and blue channel (350ex/460em). Eight images per heart were obtained for quantitative analysis. Myofibroblast percentages were derived by counting the number of single positive α-SMA cells (α-SMA+/ CD31−) and dividing by the total number of nuclei.

**Statistics and scientific rigor**. All results are presented as mean +/− SEM. All experiments were replicated at least 3 times if biological replicates were not appropriate. Statistical powering was initially performed using the nQuery Advisor 3.0 software (Statistical Solutions) along with historical data to estimate sample size. For all experiments, the calculations use $\alpha = 0.05$ and $\beta = 0.2$ (power = 0.80). Statistical analysis was performed using Prism 6.0 (GraphPad Software). Where appropriate, column analyses were performed using an unpaired, 2-tailed t-test (for 2 groups) or one-way ANOVA (for groups of 3 or more). For grouped analyses either multiple unpaired t-tests or where appropriate 2-way ANOVA with a Sidak post-hoc analysis was performed. $P$ values less than 0.05 (95% confidence interval) were considered significant. For all in vivo studies, researchers were blinded from mouse genotypes and a numerical ear tagging system enabled unbiased data collection. Upon completion of the study, mouse ID numbers were cross-referenced with genotype to permit analysis. Mice were excluded from the MI study if they lacked a scar or infarct, as evaluated by histological staining at 4 weeks post-MI.

## Data availability

The data that support the findings of this study are available from the corresponding author upon reasonable request. All ATAC-sequencing and RNA-sequencing data has been submitted to the GEO repository with accession # GSE135531.

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

## Acknowledgements

We thank Trevor Tierney for management of the lab and mutant mouse colony.

## Author contributions

Conceptualization: J.W.E.; methodology: A.A.L., T.S.L., A.A.G., B.G.H., E.M., and J.W.E.; formal analysis: A.A.L., A.A.G., P.K.L., B.G.H. and J.W.E; investigation: A.A.L, E.A., D.W.K., T.S.L., P.J., A.A.G., D.T., P.J., E.K.M.; resources: G.H., E.M., B.G.H., D.P.K., Z.P.A., K.B.M., and J.W.E; writing–original draft: A.A.L. and J.W.E; writing–review and editing: A.A.L., Z.P.A., D.P.K., K.B.M, A.A.G., and J.W.E; supervision: B.G.H., and J.W. E.; funding acquisition: A.A.L, B.G.H., and J.W.E.

## Additional information

**Competing interests:** The authors declare no competing interests.

