## [Peer Review File · Nature Communications]

Reviewers' comments:

Reviewer #1 (Remarks to the Author):

Lombardi et al present an in-depth and carefully executed study showing that decreased mitochondrial Ca uptake is an essential regulator of the fibroblast to myofibroblast transition. They develop a mechanistic causal chain showing that fibrotic stimuli such as TGF1-Beta or Angiotensin II dramatically upregulate the mitochondrial Ca uniporter complex gatekeeper protein MICU1 to suppress mCa, shift metabolism towards aerobic glycolysis, activate glutaminolysis and alpha-KG accumulation, which then decreases fibrotic gene methylation to active myofibroblast differentiation. The essential points are supported not only by the native responses to hormones, but by experiments in MCU knockout MEFs and in vivo conditional knockout of fibroblast MCU, the latter of which shows enhanced fibrogenesis after ischemia-reperfusion injury or Ang II infusion. The role of glycolysis is explored as well, by manipulating the rate controlling step at phosphofructokinase.

This work presents a convincing story and also suggests potentially novel targets that might be manipulated to alter fibrotic responses.

Although the paper is solid overall, there are a few items that could be strengthened.

- One wonders how much of the effect is simply due to the increased cytosolic Ca levels upon knockout of mitochondrial Ca uptake. It would be nice to test this by doing an intervention that blunts the increase in Ca, for example, loading BAPTA-AM into the MEFs or just culturing in lower external Ca in the media. Is there an effect related to metabolism over and above simple Ca redistribution?
- There is a global accumulation of a-KG but it is unclear whether this is due an increase in mitochondrial (IDH3, or IDH2, as in your scheme of Figure 4) or cytosolic (IDH1) activity. Typically enhanced glutaminolysis (such as occurs in cancer cells) involves increased flux through cytosolic IDH (Gln>GLU>a-KG>ICT) and then shuttling of ICT or a-KG into the mitochondria. If the TCA cycle is down regulated, perhaps the a-KG is accumulating in the cytosol because IDH1 is inhibited, possibly due to a high NADPH redox potential secondary to the increase in the pentose phosphate shunt. Can you calculate the NADPH/NADP+ ratio from your metabolome (e.g. using the a-KG/ICT ratio or another redox couple)? This might tell you something about the equilibrium of the reaction.
- Pertinent to the point above, it might be worth measuring a-KG in the mito versus cytosolic fractions with or without MCU.
- Supplemental Figure 6 is a bit sketchy since only 1 myofibroblast is shown in the entire field. A more comprehensive analysis is needed along with better examples to support Figure 6's data on myofibroblast numbers under the different conditions.

Reviewer #2 (Remarks to the Author):

The manuscript submitted by Lombardi A. et al investigates the role of mitochondrial calcium uniporter (MCU) in the context of myofibroblast differentiation. According to the authors, loss of MCU triggers a rewiring of cellular metabolism toward enhanced glycolysis. Most importantly, this leads to the activation of a myofibroblast-specific gene signature through the activity of aKG-dependent demethylases. Overall, the paper is well written, experiments are well designed and the take home message is supported by provided data. In addition, the mechanism suggested here is novel, thus representing a significant contribution that perfectly fits to this journal. However, I think that some piece of information still need further data to support the conclusion:

- The authors nicely showed that loss of MCU triggers MCU-dependent metabolic reprogramming in MEF cells. Then, they correlate this phenotype with increased MI-induced cardiac fibrosis. The authors should provide evidence that MCU-dependent metabolic reprogramming occurs also in adult cardiac fibroblasts (CF). Of course I'm not asking for the whole metabolic characterization of CFs, but they should at least demonstrate that TGFbeta/AngII modulates the MCU complex (e.g. as shown in fig 2f-k).
- It is difficult to appreciate the baseline protein levels of MICU1 in these MEF cells. When looking to

Fig.1C, MCU and MICU1 seems to be expressed at similar levels. Conversely, when looking to Fig. 2f, MICU1 looks totally absent in vehicle-treated cells (i.e. the same condition shown in fig 1c). Please provide consistent Western blots. In addition, MICU1 levels should be measured also at mRNA levels.

- TGFbeta/AngII have been reported to activate Ca²⁺ signaling in some cell types. The authors should measure cytosolic and mitochondrial Ca²⁺ dynamics induced by TGFbeta/AngII
- According to the authors, TGFbeta/AngII and/or MCU loss increase glycolysis (and consequently cellular differentiation). Although steady state metabolomics and seahorse analyses convincingly support this conclusion, it would be nice to investigate also i) glucose uptake and ii) metabolic flux analyses (e.g. using ¹³C-glucose)
- The metabolic reprogramming induced by TGFbeta and AngII are different to some extent (see fig 3). This is not surprising per se (since downstream pathways are different), but should be better discussed.

Minor points:

- Whenever possible, ratiometric Ca²⁺ probes must be preferred. Experiments in Fig 1f-g should be performed with e.g. Fura2, if you want to be truly quantitative. In addition, in fig 1d and 1f data are represented as $\Delta F/F_0$, and not F/F_0 as reported in the Y axis (F/F_0 cannot start from a 0 value, as shown in the graph)

Reviewer #3 (Remarks to the Author):

The study by Lombardi et al is interesting however it is incomplete at the current stage. Authors show in figure 5C that H3K9me3 is subject to much greater magnitude of TGFb1 dependent change (in Ad-Cre samples) than H3K27me2. Therefore, authors should carry out the experiments shown in Figure 5D, 5E and 5G for H3K9me3.

Furthermore, authors need to carry out ChIPs for H3K27me2 and H3K9me3 under the conditions shown in Figure 5P, which would ascertain the changes in these two marks when cells are treated with JIB-04.

Lombardi Nat Comm Revisions – Response to Reviewers' Critique

We'd like to thank the reviewers for their constructive critique of our work. We have tried to answer every comment/concern with new experimental data. We apologize for the delay in resubmission, the first author is a MD/PhD student and had to return to the clinic before the revision was complete which postponed our rebuttal. Also, we had technical issues with the ChIP-qPCR inhibitor experiments requested by reviewer 3 but decided to address this comment with a more robust methodology and thus performed an extensive ATAC-seq/RNA-seq study, which took us quite some time to complete and analyze. We hope our modifications to the manuscript meet your expectations.

Reviewer #1 (Remarks to the Author):

Lombardi et al present an in-depth and carefully executed study showing that decreased mitochondrial Ca uptake is an essential regulator of the fibroblast to myofibroblast transition. They develop a mechanistic causal chain showing that fibrotic stimuli such as TGF1-Beta or Angiotensin II dramatically upregulate the mitochondrial Ca uniporter complex gatekeeper protein MICU1 to suppress mCa, shift metabolism towards aerobic glycolysis, activate glutaminolysis and alpha-KG accumulation, which then decreases fibrotic gene methylation to active myofibroblast differentiation. The essential points are supported not only by the native responses to hormones, but by experiments in MCU knockout MEFs and in vivo conditional knockout of fibroblast MCU, the latter of which shows enhanced fibrogenesis after ischemia-reperfusion injury or Ang II infusion. The role of glycolysis is explored as well, by manipulating the rate-controlling step at phosphofructokinase. This work presents a convincing story and also suggests potentially novel targets that might be manipulated to alter fibrotic responses.

We thank the reviewer for their praise and constructive critique.

Although the paper is solid overall, there are a few items that could be strengthened.

1. One wonders how much of the effect is simply due to the increased cytosolic Ca levels upon knockout of mitochondrial Ca uptake. It would be nice to test this by doing an intervention that blunts the increase in Ca, for example, loading BAPTA-AM into the MEFs or just culturing in lower external Ca in the media. Is there an effect related to metabolism over and above simple Ca redistribution?

Our experiments in Figure 5i-k show that α KG alone is sufficient to promote myofibroblast differentiation independent of changes in calcium signaling. DM- α KG increased the percentage of α -SMA positive cells to the same extent as 48h of TGF β treatment (Figure 5k). These results suggest that the effect is not solely due to cytosolic increases in Ca²⁺ as no agonist was present. Further, we show that when cells are incubated with the Jumonji-C (JmjC)-domain-containing demethylase (JmjC-KDM) inhibitor, JIB-04, differentiation is blocked (Figure 5l-p).

We attempted to perform the experiment suggested using BAPTA-AM to chelate cytoplasmic calcium and separate the effect of blocking mitochondrial calcium uptake from the effect of increased cytosolic calcium, but multiple attempts were unsuccessful, because the addition of BAPTA-AM was quite toxic and often killed the cells. (We tried titrating this but could never get it to work.) However, in line with the reviewer's inquiry we do believe that alterations in mitochondrial calcium impact cytosolic calcium signaling (see Figure 1d-g) and hence can modulate cytosolic microdomains (see the slight increase in NFAT activation shown in Supplemental Figure 1). We hypothesize that modulation of the uniporter is essential for the coordinated activation of both mitochondrial and cytosolic signaling pathways to mediate cellular differentiation. We don't believe these are mutually exclusive events, and without doubt transcription factor activation is also needed for gene activation. The discussion has been reworded to reflect this ideology and our interpretation in the discussion now reflects this important

point.

“Our data suggest that extracellular stimuli are regulating cellular processes by directly altering mitochondrial signaling. We hypothesize that modulation of the uniporter is essential for the coordinated activation of both mitochondrial and cytosolic signaling pathways to mediate cellular differentiation. The outcome of this is two-fold. In addition to essential changes in mitochondrial metabolism upstream of epigenetic reprogramming, modulation of the mCa^{2+} microdomain is a way to enhance canonical cytosolic signaling pathways, hence the slight increase in NFAT activation (Supplemental Figure 1).”

2. There is a global accumulation of α -KG but it is unclear whether this is due to an increase in mitochondrial (IDH3, or IDH2, as in your scheme of Figure 4) or cytosolic (IDH1) activity. Typically enhanced glutaminolysis (such as occurs in cancer cells) involves increased flux through cytosolic IDH (Gln>GLU> α -KG>ICT) and then shuttling of ICT or α -KG into the mitochondria. If the TCA cycle is down regulated, perhaps the α -KG is accumulating in the cytosol because IDH1 is inhibited, possibly due to a high NADPH redox potential secondary to the increase in the pentose phosphate shunt. Can you calculate the NADPH/NADP+ ratio from your metabolome (e.g. using the α -KG/ICT ratio or another redox couple)? This might tell you something about the equilibrium of the reaction.

This is a good point. We calculated NADP+/NADPH redox couples from our metabolomics data to determine the equilibrium of redox reactions in our system (See new Supplemental Figure 9). These redox couples include: glucose-6-phosphate (G-6-P)/6-phosphogluconate (6-PG), 6-PG/ribulose-5-phosphate (Ru-5-P), and Glutamate (Glu)/ α -ketoglutarate (α KG) (new Supplemental Figure 5b-e). We were unable to calculate NADP+/NADPH or the isocitrate/ α KG ratios using our metabolomics data because some of these values were not obtained in our dataset. However, it's important to note that these ratios do not directly reflect directionality of the NADP+/NADPH ratio. To get a better handle on redox we performed ELISAs to measure the NADP+/NADPH ratio (new Supplemental Figure 5e). As the reviewer noted, α KG could be accumulating in the cytoplasm if IDH1 is inhibited due to a high NADPH redox potential secondary to the increase in the pentose phosphate shunt. Indeed loss of MCU results in increased NADPH (decreased NADP+/NADPH ratio shown in Supplemental Figure 5e) and this correlates with the increased flux through the pentose phosphate pathway observed in MCU knockout cells treated with TGF β (Supplemental Figure 4). However, there was no change induced by TGF β alone without MCU deletion (endogenous mechanism), suggesting that the TGF β -induced increase in α KG (Figure 4h) is not occurring via the NADP+-dependent IDH1 metabolic reaction in the cytosol. Based on our metabolomics data we hypothesized the increase in α KG is via glutaminolysis (Figure 4n) and glutaminolysis is mainly NADP+-dependent (Lee et al. 1999, Cho et al. 1995).

To further explore this hypothesis, we performed experiments to directly test the role of glutamine in myofibroblast differentiation. We inhibited glutaminolysis in WT cells by 2 mechanisms – removal of glutamine from the media and a selective inhibition of glutaminase 1 (CB-839 compound) and assessed myofibroblast differentiation by measuring

α -SMA formation by immunofluorescence (new Figure 4o-t). In both of these experiments myofibroblast differentiation was significantly attenuated, demonstrating that myofibroblast differentiation is dependent on glutamine-derived α KG.

3. Pertinent to the point above, it might be worth measuring α -KG in the mito versus cytosolic fractions with or without MCU.

We agree this is a valid inquiry, but technical limitations prevent the accurate measurement of metabolites within subcellular compartments. This is largely an issue with isolation and fractionation methods, which have scarce purity if washing steps and purification protocols are reduced or limited. Unfortunately inclusion of these crucial steps allows substantial loss of metabolites from given fractions. Even the most rapid isolation protocol reported takes ~15 minutes (Chen, et. al, Nature Protocols, 2017) and this is not sufficient to prevent metabolites from losing compartmentalization.

In addition, it remains unknown how α KG gets into the nucleus, which further hinders the ability to immobilize this particular metabolite during the fractionation procedures. While we have attempted these experiments we have been unable with any confidence to measure metabolites after fractionation, as has been communicated to us by numerous other labs we've reached out to. Therefore, we believe this experiment has several limitations and are worried about the validity of results. We are designing new tools to try and define the subcellular compartmentalization for various metabolites, such as α KG, but have not yet acquired these novel reagents. This is a focus of future projects and is an important field of study.

4. Supplemental Figure 6 is a bit sketchy since only 1 myofibroblast is shown in the entire field. A more comprehensive analysis is needed along with better examples to support Figure 6's data on myofibroblast numbers under the different conditions.

Quantification of myofibroblasts in the heart is very tricky because α SMA is also expressed in smooth muscle cells. In order to ensure we were only quantifying myofibroblasts, we excluded any α SMA+ cell that was nearby a vessel (denoted by CD31+). With this method, many myofibroblasts near blood vessels and capillaries were excluded. We counted myofibroblasts 4 weeks post-MI in the remote zone (scar and vessel formation prevented us from counting the infarcted region) and 4 weeks post-AngII infusion. After excluding all α SMA+ cells near blood vessels, there were not many myofibroblasts (α SMA+ distant from blood vessels) in each field. Myofibroblasts were quantified 4 wks post-MI in the remote zone of Col1a2-Cre mice (n=4) and *Mcu*^{fl/fl} x Col1a2-Cre mice (n=8); 3-5 sections per mouse were quantified. Myofibroblasts were also quantified in 3 Col1a2-Cre (n=3) and *Mcu*^{fl/fl} x Col1a2-Cre (n=3) Sham mice as well as 4 wks post-AngII in Col1a2-Cre mice (n=4) and *Mcu*^{fl/fl} x Col1a2-Cre (n=4) mice; 3-5 sections per mouse were quantified. Below we provided a chart with the raw data showing precisely how many heart sections/cells were counted for each mouse heart. Future follow up studies will utilize newly generated lineage tracing models to more accurately define *in vivo* fibroblast and myofibroblast number.

Injury	Genotype	Mouse #	Total # nuclei counted	Total # of α SMA+ CD31-	% myofibroblasts
4 wks post- MI (remote zone)	Col1a2-Cre	7179	1158	28	2.418%
		7189	969	56	5.779%
		7178	805	29	3.602%
		7181	1812	84	4.636%
	Mcu^{fl/fl} x Col1a2-Cre	5330	798	44	5.514%
		5877	815	47	5.767%
		5040	812	53	6.527%
		6798	1426	181	12.693%
		6796	1853	83	4.479%
		8076	2276	182	7.996%
	5340	1555	167	10.740%	
	6626	1744	79	4.530%	
Sham	Col1a2-Cre	187	1796	22	1.225%
		9190	1865	16	0.858%
		37	2355	26	1.104%
	Mcu^{fl/fl} x Col1a2-Cre	161	2514	51	2.029%
		349	2312	24	1.038%
		350	1910	36	1.885%
4 wks-post AngII	Col1a2-Cre	1	2391	45	1.882%
		189	2378	39	1.640%
		193	1635	32	1.957%
		190	1895	30	1.583%
	Mcu^{fl/fl} x Col1a2-Cre	47	2449	77	3.144%
		52	2134	51	2.390%
		9999	2630	75	2.852%
		9996	1239	31	2.502%

Reviewer #2 (Remarks to the Author):

The manuscript submitted by Lombardi A. et al investigates the role of mitochondrial calcium uniporter (MCU) in the context of myofibroblast differentiation. According to the authors, loss of MCU triggers a rewiring of cellular metabolism toward enhanced glycolysis. Most importantly, this leads to the activation of a myofibroblast-specific gene signature through the activity of α KG-dependent demethylases. Overall, the paper is well written, experiments are well designed and the take home message is supported by provided data. In addition, the mechanism suggested here is novel, thus representing a significant contribution that perfectly fits to this journal.

We thank the reviewer for their positive appraisal of our work and insightful comments.

1. The authors nicely showed that loss of MCU triggers MCU-dependent metabolic reprogramming in MEF cells. Then, they correlate this phenotype with increased MI-induced cardiac fibrosis. The authors should provide evidence that MCU-dependent metabolic reprogramming occurs also in adult cardiac fibroblasts (CF). Of course I'm not asking for the whole metabolic characterization of CFs, but they should at least demonstrate that TGFbeta/AngII modulates the MCU complex (e.g. as shown in fig 2f-k).

Our *in vivo* data in Figure 6 is the most robust data to support this does occur in cardiac fibroblasts in a MCU-dependent fashion. Loss of MCU in fibroblasts (*Mcu^{f/f}* x *Col1a2-Cre*) exacerbates fibrotic remodeling post-MI and in the chronic Angiotensin II infusion model. We observed a significant increase in myofibroblast formation with loss of fibroblast MCU post-MI and AngII. Additionally, there was a trend towards an increase in myofibroblast differentiation in sham *Mcu^{f/f}* x *Col1a2-Cre* mice. To further support this data and our hypothesis that fibrotic stimuli signal to downregulate mitochondrial Ca^{2+} uptake by upregulation of MICU1 (as shown in Figure 2), we isolated mouse adult cardiac fibroblasts, as requested, and treated them with TGF β or AngII followed by qPCR analysis of MICU1 expression. This data supports that fibrotic agonists acutely upregulate MICU1 expression in cardiac fibroblasts (new Figure 2 l,q and new Supplemental Figure 2 p-s) and that this is a rapid transcriptional response.

2. It is difficult to appreciate the baseline protein levels of MICU1 in these MEF cells. When looking to Fig.1C, MCU and MICU1 seems to be expressed at similar levels. Conversely, when looking to Fig. 2f, MICU1 looks totally absent in vehicle-treated cells (i.e. the same condition shown in fig 1c). Please provide consistent Western blots. In addition, MICU1 levels should be measured also at mRNA levels.

We agree with the reviewer that it is difficult to appreciate the baseline levels of MICU1 in the blots in Figure 2. This is because of the substantial increase in MICU1 expression that occurs just 12h after TGF β or AngII. Overexposure of the same western blots allows for visualization of the MICU1 band at baseline is shown below.

Higher exposure MEFs + TGF β :

Higher exposure MEFs + AngII:

In addition, we analyzed MICU1 expression by qPCR after both TGF β and AngII treatment and also found a similar increase in expression. We have added this qPCR data to Figure 2 (new Figure 2k, p and new Supplemental Figure 2I-o).

3. TGF β and AngII have been reported to activate Ca²⁺ signaling in some cell types. The authors should measure cytosolic and mitochondrial Ca²⁺ dynamics induced by TGF β /AngII.

To corroborate our data in Figure 2 showing that TGF β decreases mitochondrial calcium uptake, we used live cell imaging to examine cytosolic and mitochondrial calcium dynamics in vehicle and TGF β -treated cells. Cells were transduced with adenovirus encoding a mitochondrial-targeted genetic Ca²⁺ reporter (Mito-R-GECO) for 48h. Prior to live-cell imaging, cells were loaded with the calcium sensitive dye Fluo-4 AM to measure cytosolic calcium (_cCa²⁺) transients. After baseline recordings, cells were treated with ATP to initiate purinergic receptor-mediated IP3R Ca²⁺ release. Cells that were pretreated with TGF β for 12h showed significantly increased _cCa²⁺ and significantly decreased _mCa²⁺ load (new Figure 2a,b and new Supplementary Figure 2a,b). These data support our hypothesis that fibrotic agonists downregulate _mCa²⁺ uptake to initiate downstream fibrotic signaling and myofibroblast differentiation (Figure 7).

4. According to the authors, TGF β /AngII and/or MCU loss increase glycolysis (and consequently

cellular differentiation). Although steady state metabolomics and seahorse analyses convincingly support this conclusion, it would be nice to investigate also glucose uptake and ii) metabolic flux analyses (e.g. using ^{13}C -glucose).

In addition to the steady state metabolomics and seahorse data, we also demonstrated the role of glycolysis by modifying one of the rate limiting enzymes in glycolysis using adenovirus-encoding mutants of PFK2 in order to enhance or diminish glycolysis (Figure 3n-b'). These data are causal in nature and go beyond showing changes in uptake or flux.

To further demonstrate that the loss of MCU enhances glycolysis we performed a glucose-uptake assay, as requested, which showed that loss of MCU or treatment with TGF β significantly increases glucose uptake. Future experiments will examine changes in glucose flux using SIRM given the extensive nature of these experiments and will be reported in a follow up manuscript.

5. The metabolic reprogramming induced by TGFbeta and AngII are different to some extent (see fig 3). This is not surprising per se (since downstream pathways are different), but should be better discussed.

We hypothesize that the upregulation of MICU1 promotes aerobic glycolysis, which leads to downstream metabolic and epigenetic changes driving myofibroblast differentiation. Consistent with their well-known role in myofibroblast differentiation, both AngII and TGF β significantly upregulate MICU1 and glycolysis. As the reviewer points out, we observed a transient increase in glycolysis with AngII treatment in contrast to an increase in both glycolysis and oxidative phosphorylation with TGF β treatment. We hypothesize that both of these stimuli are important for the metabolic switch that drives fibroblast-to-myofibroblast differentiation (via an early increase glycolysis), while TGF β is also responsible for maintaining higher metabolic activity (sustained increases in glycolysis in addition to increased OxPhos) in the fully differentiated myofibroblast in order to meet the high demands of this much larger, synthetic, contractile cell. Further, previous reports suggest that AngII signaling upregulates expression and activation of TGF β . Therefore, another possibility is that early increases in glycolysis mediated by AngII activate myofibroblast differentiation and expression of TGF β (reviewed in Weber, et. al, Nature Reviews Cardiology, 2013). Subsequently, TGF β expression further promotes myofibroblast differentiation and activity. We have added this to our interpretation of results.

Minor points:

6. Whenever possible, ratiometric Ca²⁺ probes must be preferred. Experiments in Fig 1f-g should be performed with e.g. Fura2, if you want to be truly quantitative. In addition, in fig 1d and 1f data are represented as deltaF/F₀, and not F/F₀ as reported in the Y axis (F/F₀ cannot start from a 0 value, as shown in the graph).

Thank you for identifying this; the y-axis has been changed for these graphs in Figure 1d and 1f.

Reviewer #3 (Remarks to the Author):

We thank you for your insightful comments and critique of our work.

1. The study by Lombardi et al is interesting however it is incomplete at the current stage. Authors show in figure 5C that H3K9me3 is subject to much greater magnitude of TGFβ1 dependent change (in Ad-Cre samples) than H3K27me2. Therefore, authors should carry out the experiments shown in Figure 5D, 5E and 5G for H3K9me3.

While we did see some changes in H3K9me3 (data below), but the data suggests these changes may not be dependent on alterations in mitochondrial Ca²⁺ uptake since the TGFβ-mediated pattern of changes seen in the control cells is different than that in the *Mcu* knockout cells (see quantification below). Therefore, we focused on H3K27me2 because we hypothesized these changes are mediated by alterations in mitochondrial calcium uptake at the level of the uniporter. Quantification of the other methylation marks is now presented in new Supplemental Figure 6. Future studies are planned employing non-biased global histone proteomics to quantitatively examine all possible histone PTMs during myofibroblast formation.

2. Furthermore, authors need to carry out ChIPs for H3K27me2 and H3K9me3 under the conditions shown in Figure 5P, which would ascertain the changes in these two marks when cells are treated with JIB-04.

Unfortunately we found that treatment with JIB-04 interfered with our ability to reliably pull down chromatin with our H3K27me2 antibody. We tried to trouble shoot this numerous times but were unable to get this experiment to work. However, we completely agree with the reviewer that true evidence of chromatin remodeling in our model system is needed beyond the ChIP-qPCR data, which correlates accessibility with changes in a given PTM at a single small loci. To better address this, we employed ATAC-seq in conjunction with RNA-seq on fibroblasts treated with vehicle or TGFβ. This

methodology allowed the global assessment of ‘chromatin accessibility’ at the regulatory regions of important myfibroblast genes with determination of actual mRNA levels in the same samples. These experiments took us months to perform and analyze and we hope the reviewer agrees that this is superior to ChIP and is satisfactory for the current manuscript.

We found that TGFβ induced significant changes in chromatin structure and greatly enhanced accessibility at key myfibroblast genes (data shown here and following page for *Postn*, *Acta2*, *Col1a1*, *Lox*, *Wnt1* and *Tgfb* – see new Figures 5g and Supp. Figures 6h-I). Importantly, ATAC-derived increases in open chromatin structure correlated with increased or decreased mRNA transcription in a directionality in agreement for what’s been reported for myfibroblast formation (*Kanisicak et al. Nat Comms 2016*). This data shows that acute TGFβ signaling modifies chromatin structure, which we hypothesize is through demethylation of H3K27me2, in order to permit changes in gene transcription that are essential for myfibroblast differentiation.

Supplemental Figure 7

REVIEWERS' COMMENTS:

Reviewer #1 (Remarks to the Author):

The authors have exceeded all expectations in replying to the concerns raised. Very nice work.

The data on redox couples in Suppl Fig 5 is fairly convincing that mitochondrial NADPH redox (inferred from the Glu:a-KG) is actually much more impacted by MCU-KO than cytosolic NADPH (G6P:6PG, which tends to go in the opposite direction). You might want to make that conclusion clear in the legend of Suppl fig 5.

Kudos for adding the ATAC seq data.

Reviewer #2 (Remarks to the Author):

The authors successfully addressed my original comment. I think this manuscript now deserves publication.

Reviewer #3 (Remarks to the Author):

Authors have answered reviewer's questions in a satisfactory manner, thank you!

RESPONSE TO REVIEWERS' COMMENTS:

Reviewer #1 (Remarks to the Author):

The authors have exceeded all expectations in replying to the concerns raised. Very nice work.

Thank you for your praise.

The data on redox couples in Suppl Fig 5 is fairly convincing that mitochondrial NADPH redox (inferred from the Glu:a-KG) is actually much more impacted by MCU-KO than cytosolic NADPH (G6P:6PG, which tends to go in the opposite direction). You might want to make that conclusion clear in the legend of Suppl fig 5.

We have added a comment in the Supplemental Figure 5 legend as suggested.

Kudos for adding the ATAC seq data.

Thanks!

Reviewer #2 (Remarks to the Author):

The authors successfully addressed my original comment. I think this manuscript now deserves publication.

Thank you for your positive appraisal of our work.

Reviewer #3 (Remarks to the Author):

Authors have answered reviewer's questions in a satisfactory manner, thank you!

Thank you for your positive appraisal of our work.